# Dynamic partitioning of branched-chain amino acids-derived nitrogen supports renal cancer progression

Marco Sciacovelli[1,2,18], Aurelien Dugourd[3,4,18], Lorea Valcarcel Jimenez [1,5], Ming Yang[1,5], Efterpi Nikitopoulou[1], Ana S. H. Costa [1,6], Laura Tronci[1], Veronica Caraffini[1], Paulo Rodrigues[1], Christina Schmidt[1,5], Dylan Gerard Ryan [1], Timothy Young[1], Vincent R. Zecchini[1], Sabrina H. Rossi [7], Charlie Massie [7], Caroline Lohoff[3], Maria Masid [8,9], Vassily Hatzimanikatis [8], Christoph Kuppe [4,10], Alex Von Kriegsheim[11], Rafael Kramann[4,10,12], Vincent Gnanapragasam [13], Anne Y. Warren[14], Grant D. Stewart [13], Ayelet Erez [15], Sakari Vanharanta [1,16,17], Julio Saez-Rodriguez [3,19] ✉ & Christian Frezza [1,5,19] ✉

Metabolic reprogramming is critical for tumor initiation and progression. However, the exact impact of specific metabolic changes on cancer progression is poorly understood. Here, we integrate multimodal analyses of primary and metastatic clonally-related clear cell renal cancer cells (ccRCC) grown in physiological media to identify key stage-specific metabolic vulnerabilities. We show that a *VHL* loss-dependent reprogramming of branched-chain amino acid catabolism sustains the de novo biosynthesis of aspartate and arginine enabling tumor cells with the flexibility of partitioning the nitrogen of the amino acids depending on their needs. Importantly, we identify the epigenetic reactivation of argininosuccinate synthase (ASS1), a urea cycle enzyme suppressed in primary ccRCC, as a crucial event for metastatic renal cancer cells to acquire the capability to generate arginine, invade in vitro and metastasize in vivo. Overall, our study uncovers a mechanism of metabolic flexibility occurring during ccRCC progression, paving the way for the development of novel stage-specific therapies.

Cancer is an ever-evolving disease in which tumor cells are subject to constant changes in nutrient and oxygen availability within the tumor microenvironment. To adapt to different microenvironments during tumor evolution, cancer cells become metabolically flexible, a process orchestrated either directly by metabolites availability or by activation of oncogenic signaling[1]. Consistently, it has been shown that tumors at different stages are metabolically distinct[2–7]. For instance, solid tumors use nutrients such as glucose to generate the biomass necessary to sustain their high proliferative demands[4,7], whereas successful metastasis relies more on pyruvate, glutamine, lipid metabolism and, in

specific tumor types, on mitochondrial metabolism such as oxidative phosphorylation[6,7]

High-throughput metabolomics technologies are widely used to study cancer metabolism. However, despite the simultaneous measurement of hundreds of metabolites, this approach cannot fully capture the complexity and dynamics of the altered metabolic network. Therefore, it is crucial to develop computational algorithms that can extract more biological insight from sparse metabolomics data[8–10]. These methods, combined with in vitro experimental conditions that mimic the nutrient microenvironment of the tumor in vivo[11,12], can be

used not only to dissect the complexity of tumor metabolism regulation, but also to identify new metabolic vulnerabilities in vivo.

Clear cell renal cell carcinoma (ccRCC), the most common histological subtype of RCC that accounts for 70% of renal malignancies[13], has been extensively studied for its profound metabolic reprogramming[14–16]. ccRCC arises from epithelial tubular cells[17] and it is driven by (epi)genetic lesions affecting the Von Hippel-Lindau tumor suppressor (*VHL*). *VHL* loss leads to robust activation of pro-oncogenic signaling mediated by the hypoxia inducible factor 2A (*EPAS1*/ HIF2A)[18–20], which transcriptionally orchestrates some of the most prominent metabolic alterations of these tumors. ccRCC tumors are fueled by aerobic glycolysis rather than oxidative phosphorylation (OXPHOS) due to HIF-mediated metabolic reprogramming and the mitochondrial dysfunction frequently observed in these tumors[14,21,22]. Over the last years, other pathways were added to the metabolic landscape of ccRCC, including dysregulated tryptophan, arginine, and glutamine metabolism, together with enhanced lipid and GSH biosynthesis[14]. Only recently, it was shown that the genomic loss or suppression of urea cycle (UC) genes such as Arginase 2 (*ARG2*) and argininosuccinate synthetase (*ASS1*) favors renal cancer growth, preserving the consumption of pyridoxal 5′-phosphate[23]. ccRCC tumors are also metabolically flexible, and the metabolic landscape of late-stage renal cancers is distinct from that of primary renal tumors[16]. More specifically, it was shown that upregulation of GSH biosynthesis, cysteine/methionine metabolism and polyamine pathways is associated with advanced ccRCC[16]. However, how the metabolic landscape of renal tumors evolves through progression, is regulated at molecular level, and impacts on tumor biology is largely unknown.

In this work, through a multi-omic analysis of primary and metastatic ccRCC cells, we identify the reprogramming of the branched-chain amino acid catabolism in renal cancer cells as a source of metabolic flexibility that sustains tumor growth. We showed that this reprogramming, dependent on *VHL* loss, provides cancer cells with enhanced capability to synthetize aspartate using the nitrogen derived from BCAA. Importantly, we observed that additional epigenetic reactivation of *ASS1* specifically in metastatic cells allows them to generate arginine using the nitrogen of BCAA, invade in vitro and metastasize in vivo. Overall, our study identify a mechanism of metabolic flexibility occurring during ccRCC progression that has the potential to become a target for future stage-specific intervention strategies.

## Results

To identify metabolic pathways reprogrammed in ccRCC progression, we first performed an enrichment analysis (GSEA) of tumor vs. matched normal tissue using the ccRCC(KIRC) RNA-seq dataset from The Cancer Genome Atlas (TCGA) (Fig. 1a). We identified amongst the most upregulated pathways in the tumors ribosome, DNA replication and signaling cascades while key metabolic features dysregulated in ccRCC tumors included not only the suppression of OXPHOS and TCA cycle but also arginine, BCAAs, tryptophan, and pyruvate metabolism (Fig. 1a), in line with previous findings[16,19,24–26]. BCAA catabolism (valine, leucine and isoleucine degradation) was the most suppressed pathway in renal tumors. Interestingly, all the genes of the pathway were significantly downregulated in the tumor samples, except for the Branched-Chain Amino Acid Transaminase 1 (*BCAT1*) and the lysosomal amino acids oxidase Interleukin 4 Induced 1 (*IL4I1*), which were strongly upregulated (Fig. 1b). This apparent discrepancy between the expression of BCAT1 and the other genes from the BCAA catabolism suggests that additional mechanisms beyond the transcriptional control may be involved in the fine-tuning of the pathways in tumors. Then, we focused on metabolic pathways that are transcriptionally deregulated during ccRCC progression. To this end, we compared RNA-seq from patients with locally advanced and metastatic (stage III + IV) vs. localized (stage I + II) ccRCC tumors. Using this approach,

we identified metabolic pathways suppressed in stage III-IV cancers, with the BCAA catabolism as the top downregulated one (Fig. 1c). Consistent with a role for BCAA catabolism in ccRCC progression, the overall survival of patients with ccRCC correlated with the expression level of this pathway (Fig. 1d), with high expression associated with better prognosis. Of note, a significant correlation between BCAA enzyme levels and patient survival was only observed in a few tumor types, including renal cancer (KIRC and kidney renal papillary cell carcinoma KIRP) and colorectal cancer (Supplementary Fig. 1a, b). Thus, the suppression of BCAA catabolism is a metabolic hallmark of renal cancer and is independent of the cancer stage.

To assess the role of BCAA catabolism in renal cancer, we compared HK-2 proximal tubule kidney epithelial cells with a panel of ccRCC cell lines, 786-O, OS-RC-2, RFX-631 and the metastatic derivatives, 786-M1A, 786-M2A, and OS-LM1[27] (Fig. 2a). To mimic the nutrient availability in vivo, we cultured all the cell lines in Plasmax, a recently developed physiological medium based on the human serum's nutrient composition[12]. First, we performed a liquid chromatography-mass spectrometry (LC–MS) metabolomic analysis of the cells stably grown in Plasmax or standard culture medium (RPMI) and correlated it with the metabolic profile of a cohort of renal tumors and matched healthy renal tissues[9]. The metabolic profiles of cells grown in Plasmax exhibited a significant correlation with the profiles of tumor and normal tissues ($p$-value $< 10^{-6}$), which is slightly higher compared to that of cells cultured in RPMI ($p$-value $< 10^{-8}$) (Supplementary Fig. 2a). Furthermore, when we analyzed their transcriptomic profile (Supplementary Fig. 3g–j), ccRCC cells grown in Plasmax displayed the activation of transcription factors (TF) such as Hypoxia-inducible Factor 2A (HIF2A, *EPAS1* gene), MYC Associated Factor X (*MAX*), and Paired Box 8 (*PAX8*), known drivers of ccRCC[20,27,28] (Supplementary Fig. 2b). Consistent with previous data[27,29], the metastatic cell lines maintained the expression of specific metastatic markers such as the C-X-C Motif Chemokine Receptor 4 (*CXCR4*), the Cytohesin 1 Interacting Protein (*CYTIP*), the Latent Transforming Growth Factor Beta Binding Protein 1 (*LTBP1*) and the SLAM Family Member 8 (*SLAMF8*) (Supplementary Fig. 2c). We then investigated the differential expression of the metabolic pathways in the renal cells HK2, 786-O and 786-M1A cultured in Plasmax using proteomics (Supplementary Fig. 3d–f). Enrichment analysis of proteomics data indicated that glycolysis, purine, and glutathione metabolism were upregulated while the BCAA catabolism, together with the OXPHOS and the TCA cycle, were amongst the most suppressed metabolic pathways in both 786-O and 786-M1A when compared with HK2 cells (Fig. 2b), in line with the results of the renal tumors from patients (Fig. 1a). Importantly, the majority of the proteins detected that belong to the BCAA catabolism were suppressed in 786-O vs. HK2 cells with the exception of a few enzymes including BCAT1, Short/Branched-Chain Specific Acyl-CoA Dehydrogenase (ACADSB), and the Aldehyde Dehydrogenase 2 (ALDH2) (Supplementary Fig. 2d). Notably, the levels of both BCAA catabolism and OXPHOS related proteins were further suppressed in metastatic 786-M1A compared to primary 786-O (Fig. 2c), as observed in the most aggressive renal tumors (Fig. 1c). The suppression of the OXPHOS in all renal cancer cells was confirmed by the lower basal and stimulated cellular respiration compared to HK2 cells (Supplementary Fig. 2e). To functionally validate the GSEA results, we used our metabolomics data (Fig. 2d, e and Supplementary Fig. 3a–c). We observed that both 786-O and 786-M1A have lower intracellular levels of leucine and isoleucine, while they accumulated C5 carnitines and methylmalonylcarnitines (detected as methylmalonylcarnitine+succinylcarnitine), by-product metabolites derived from intermediates of BCAA catabolism, while no significant differences were observed in C3-carnitines (Fig. 2d–f and Supplementary Fig. 2f). The accumulation of by-product metabolites derived from intermediates of BCAA catabolism might be the consequence of the suppression of key acyl-CoA dehydrogenases that belong to the BCAA catabolism such as Isovaleryl-CoA dehydrogenase

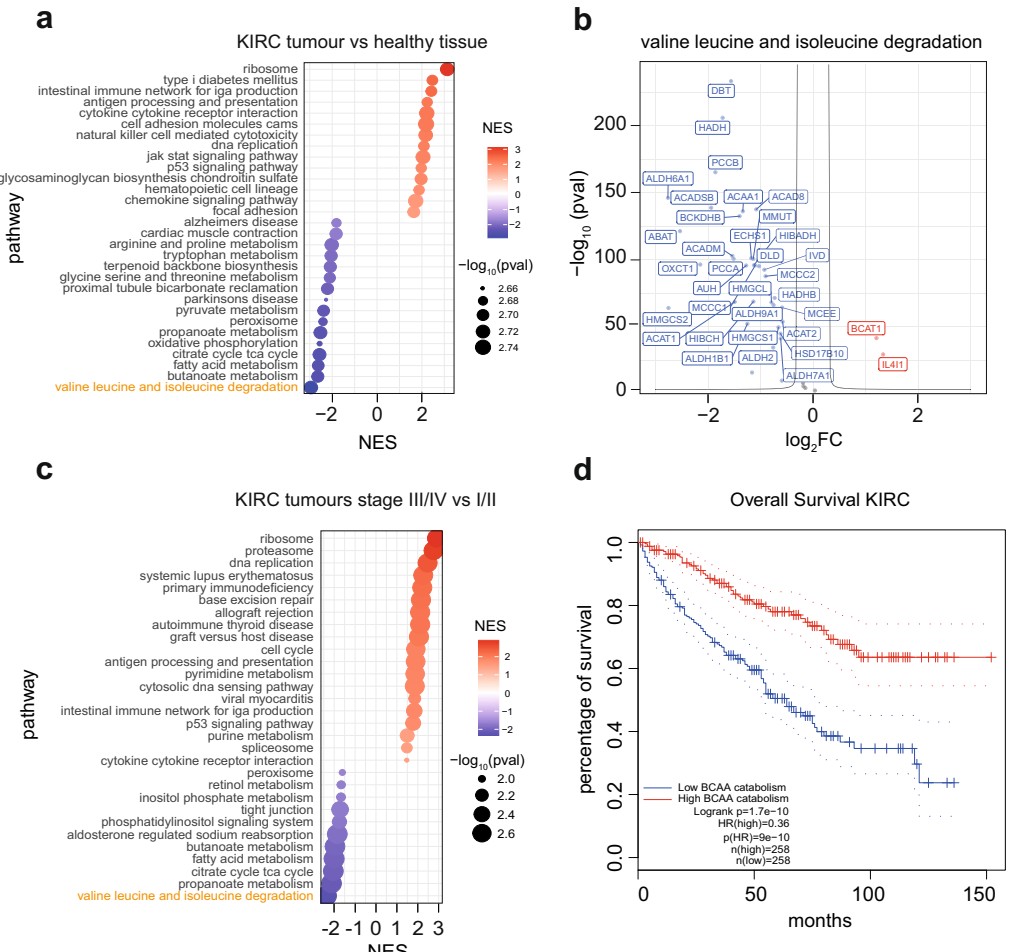

**Fig. 1 | Branched-chain amino acid catabolism is suppressed in KIRC. a** Dot plot showing the enriched pathways ranked by significance in KIRC tumors compared to renal healthy tissue obtained through GSEA analysis of RNA-seq data from TCGA. The dot size represents the significance expressed as $-\log_{10}(p\text{-value})$. red dots = upregulated pathways, blue dots = downregulated pathways. NES normalized enrichment score. Statistics was calculated using moderated two-sided Student test, $p$-values were corrected with Benjamini−Hochberg procedure. **b** Volcano plot showing the differential expression of genes that belong to KEGG "Valine leucine and isoleucine degradation" signature in KIRC tumors compared to renal healthy tissue. FC fold-change, red = upregulated genes, blue = downregulated genes. **c** Dot plot of the differentially enriched pathways in KIRC tumors comparing stage III/IV vs. stage I/II. Pathways, ranked by significance, are obtained through GSEA analysis of TCGA RNA-seq data. The dot size represents the significance expressed as $-\log_{10}(p\text{-value})$. red dots = upregulated pathways, blue dots = downregulated pathways. NES normalized enrichment score. Statistics was calculated using moderated two-sided Student test, $p$-values were corrected with Benjamini−Hochberg procedure. **d** Overall survival of KIRC patients obtained through GEPIA, based on gene expression of KEGG 'Valine, leucine and isoleucine degradation' signature. Cut-off used for high/low groups was 50% and $p$-value displayed as $-\log\text{rank}(p\text{-value})$ calculated using Mantel−Cox test. The dotted line refers to the survival with a confidence interval (CI) of 95%. $n$ number of samples compared, HR = hazard ratio based on the Cox PH model. KIRC = Renal clear cell carcinoma.

(IVD) and Methylmalonyl-CoA mutase (MUT) (Fig. 2g). Intriguingly, metastatic 786-M1A cells displayed higher accumulation of methyl-malonylcarnitine when compared to 786-O cells (Fig. 2e and Supplementary Fig. 2f), suggesting a potentially enhanced dysregulation of BCAA catabolism. Finally, the uptake of BCAAs was not significantly different among the cells types, with the exception of valine in 786-O and 786-M1A cells (Supplementary Fig. 2g), even though the hetero-dimer transport system between the Solute Carrier Family 7 Member 5 (SLC7A5, LAT1) and Solute Carrier Family 3 Member 2 (SLC3A2, CD98) (Supplementary Fig. 2h) was upregulated in all ccRCC cells compared to HK2 cells.

To further characterize the metabolic landscape of ccRCC cells during progression, including the reprogramming of BCAA catabolism, we developed ocEAn (metabOliC Enrichment Analysis), a computational method that generates a metabolic footprint for each metabolic enzyme present in the recon2 metabolic reaction network[30] (Supplementary Fig. 4a, methods), to our metabolomics data. These footprints show the metabolites directly or indirectly associated with a given metabolic enzyme, their abundances and relative position either upstream or downstream of the reaction. Through the metabolic footprints, ocEAn provides an overview of the metabolic alterations centered on the single enzyme, highlighting patterns of imbalance between the upstream and downstream metabolites mapped in the enzyme footprint (Supplementary Fig. 4a; full interactive network available at: https://sciacovelli2021.omnipathdb.org). We applied this tool to study the activity of BCAT1, a key enzyme at the entry point of BCAA catabolism in our renal cellular models. BCAT1 was found upregulated both in the tumors from TCGA (Fig. 1b) and in ccRCC cells at the protein level (Supplementary Fig. 2d). All ccRCC cells displayed lower BCAAs levels upstream of BCAT1 with a significant upregulation of carnitines derived from intermediates of the BCAA catabolism, notably methylmalonylcarnitine (log2 FC > 2, FDR < 10$^{-40}$) and C5-carnitines downstream of the reaction (Fig. 3a and Supplementary Fig. 2f). One of the benefits of ocEAn is the possibility to uncover deregulated metabolites indirectly associated with an enzyme, either upstream or downstream, that might contribute to its biological function. Intriguingly, we found that argininosuccinate, an intermediate product of the urea cycle strongly downregulated in 786-O

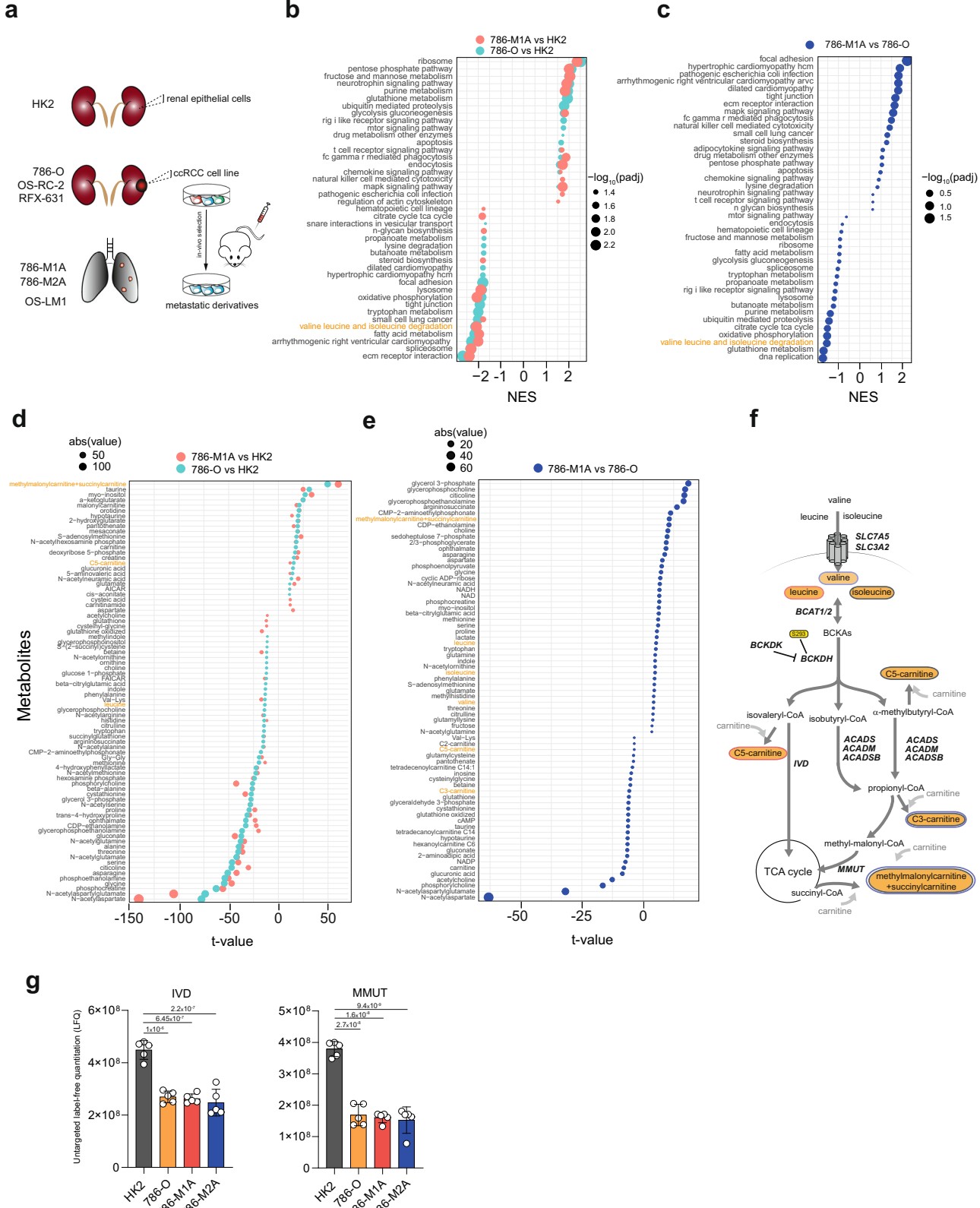

compared to HK2, was the top upregulated metabolite downstream of BCAT1 in the metastatic 786-M1A cells when compared to 786-O (Fig. 3a). This result suggests that some products of BCAT transamination are shunted in the urea cycle in the renal cancer cells and also that the functioning of the urea cycle might differ between 786-M1A vs. 786-O.

To understand the biological relevance of the BCAA catabolism reprogramming in ccRCC progression and its involvement in the acyl-carnitines accumulation, we cultured HK2, 786-O and the derived metastatic cells (786-M1A and 786-M2A) together with additional ccRCC cells (OS-RC-2 and metastatic derivatives OS-LM1; RFX-631, Fig. 2a) in the presence of $^{13}C_6$ leucine+isoleucine and we measured the

**Fig. 2 | Regulation of BCAA catabolism in a cellular model system for renal cancer progression. a** Schematics of the cell lines used in the study. HK2 cells were derived from normal renal tissue, 786-O, OS-RC-2, RFX-631 from primary ccRCC and the metastatic 786-M1A, 786-M2A, and OS-LM1 from lung metastases after injection in vivo. **b, c** Dot plot of the enriched pathways ranked by significance obtained through GSEA from proteomic data. Green dots represent 786-O vs. HK2, orange 786-M1A vs. HK2, blue 786-M1A vs. 786-O. Statistics was calculated using moderated two-sided Student test, *p*-values were corrected with Benjamini–Hochberg procedure. Dot size is proportional to $-\log_{10}$(adj-*p*-value). NES normalized enrichment score. **d, e** Dot plot showing the differential abundance of the indicated intracellular metabolites in the comparisons 786-O vs. HK2 (green),786-M1A vs. HK2 (orange), and 786-M1A vs. 786-O (blue) ranked by *t*-values. Data were normalized to total ion count and generated from *N* = 3 experiments. The dimension of the dots represents abs *t*-values. **f** Simplified schematic of the BCAA catabolism. Leucine and isoleucine are imported by the solute carrier system SLC7A5/SLC3A2, converted into branched-chain keto acids (BCKAs) by BCAT1/2 and subsequently oxidized by BCKDH complex into acyl-CoAs. BCKDH complex is inhibited by BCKDK-dependent phosphorylation on Ser293 residue. C5 and C3-carnitines are measured as readout of isovaleryl-CoA and propionyl-CoA, respectively. Acyl-CoAs are further catabolized by IVD, ACADS, ACADSB, ACADM, MMUT before entering the TCA cycle. Methylmalonylcarnitine and succinylcarnitine are readouts of methylmalonyl-CoA and succinyl-CoA. Metabolites highlighted by orange circles are measured by LC–MS. Red circles = metabolites from leucine catabolism, black circles = metabolites derived from isoleucine, blue circles = metabolites derived from valine. **g** Labeled-free quantification (LFQ) of the indicated proteins from the proteomics dataset. Data are shown as mean of 5 independent cultures ± SD. Significance was calculated using one-way ANOVA where each group was compared with HK2. SLC7A5 = solute carrier family 7 member 5; SLC3A2 solute carrier family 3 member 2, BCAT1/2 branched-chain amino acid transaminase 1/2, BCKDH branched-chain keto acid dehydrogenase complex, BCKAs branched-chain keto acids, BCDK branched-chain keto acid dehydrogenase kinase, IVD isovaleryl-CoA dehydrogenase, ACADS Acyl-CoA dehydrogenase short chain, ACADSB Acyl-CoA dehydrogenase short/branched-chain, ACADM Acyl-CoA dehydrogenase medium chain, MMUT methylmalonyl-CoA mutase.

generation of labeled downstream metabolites, including KIC/KMV, C5 and C3-carnitines, and TCA cycle intermediates fumarate and malate (Supplementary Fig. 4b). We detected higher labeling in both C5 carnitines and C3-carnitines (C5-carnitine m+5 and C3 carnitine m+3) in all cancer cells at 1 h and 3 h time points compared to HK2 (Fig. 3b) while leucine+isoleucine or KIC + KMV labeled percentages were similar among all cells. These results showed that the upper part of the BCAA catabolism is still functional in all renal cancer cells, even more active in ccRCC than normal HK2, independently from the tumor stage. However, the full oxidation of BCAAs did not significantly contribute to the generation of TCA cycle intermediates in all renal cells since we detected a very low fraction of labeled fumarate (3%) and malate (below 1%) (Fig. 3b). We incubated HK2, 786-O and the derived metastatic cells with the $^{13}C_6$ leucine+isoleucine for a longer time (43 h), but similarly to the shorter time points, the labeled percentages of both C2-carnitines and fumarate were below 1% despite comparable levels of labeled intracellular leucine (Supplementary Fig. 4c). Whilst the very low percentage of labeled C2-carnitine might be due to the presence of C2-carnitine in the medium, these results are consistent with previous reports that showed a limited contribution of BCAAs oxidation in the TCA cycle in vivo in the kidneys[31] and other tumor types[32,33].

The BCAA catabolism represents an important source of nitrogen for amino acids synthesis, based on the production of glutamate through transamination of BCAA by BCATs[33,34]. We have shown that the BCAAs are not significantly contributing to the generation of TCA cycle intermediates in all the renal cells used (Fig. 3b and Supplementary Fig. 4c) and moreover, ocEAn highlighted the significant dysregulation of aspartate, asparagine and argininosuccinate downstream of BCAT1 (Fig. 3a). Therefore, we hypothesized that the reprogramming we observed in ccRCC might provide a nitrogen source for the generation of glutamate and other downstream metabolic reactions. To experimentally validate the biological role of BCAT transamination we cultured HK2, 786-O and 786-derived metastatic cells in the presence of $^{15}N$ leucine+isoleucine in Plasmax and measured the generation of $^{15}N$-labeled glutamate (Fig. 4a). Glutamate is a key amino acid used in multiple metabolic pathways. For instance, it donates the nitrogen for the conversion of oxaloacetate into aspartate catalyzed by Glutamate Oxaloacetate Transaminases (GOT1/GOT2), which through asparagine synthase (ASNS) can be in the end converted into asparagine (Fig. 4a). We found that all ccRCC cells generated significantly more glutamate, aspartate, and asparagine labeled from leucine and isoleucine (Fig. 4b, c and Supplementary Fig. 5a). Among other glutamate-derived amino acids, we also detected increased labeling in proline (proline m+1) in 786-O, 786-M1A and 786-M2A cells, while serine, glutamine and alanine m+1 were lower than in HK2 cells (Supplementary Fig. 5a). To derive aspartate from leucine, cancer cells rely on the reverse reaction of GOTs, which consumes glutamate derived from leucine

transamination and OAA to generate αKG and aspartate[35]. In line with this observation, GOT1 protein levels were higher in ccRCC cells, while on the contrary, GOT2 was suppressed (Supplementary Fig. 5b). Similarly, we also detected an increase in ASNS protein levels in all renal cancer cells, in line with the increased labeling of asparagine in ccRCC (Supplementary Fig. 5b). Of note, a similar metabolic rewiring was observed in OS-RC-2, OS-LM1 and RFX-631 cells, that derived a higher amount of glutamate, aspartate (with the exception of RFX-631 cells), and asparagine from BCAA when compared to normal HK2 cells (Supplementary Fig. 5c).

To better understand the maximal contribution of BCAT transamination to the generation of aspartate, we cultured the cells in EBSS where exogenous aspartate, glutamate and other amino acids are absent, with the exception of $^{15}N$ leucine. Strikingly, the net contribution of BCAT transamination to de novo generation of aspartate in these conditions reaches more than 60% in renal cancer cells (Fig. 4d), while in HK2 cells it is below 20% even though the intracellular percentage of labeled leucine in all cell types is comparable (Supplementary Fig. 5d). We did not observe differences in the relative percentage of the labeled glutamate in these conditions among the cells. Considering that aspartate is limiting for nucleotide biosynthesis[36,37], we then assessed whether BCAT1 activity indirectly contributes to nucleotide pools. To test this hypothesis, we suppressed BCAT activity with a pharmacological inhibitor (BCAT inhibitor 2, BCATI, Fig. 4a), which preferentially targets the cytosolic BCAT1 isoform[38]. As a result of the BCAT inhibition, we observed a suppression of both the BCKAs (KIC+KMV) downstream of BCAT in all cell types, together with C5-carnitines (Supplementary Fig. 5e). Importantly, the inhibition of the transamination also significantly affected intracellular glutamate and aspartate levels, even though the latter mainly occurred in metastatic cells (Fig. 4e). As a consequence of the alterations of the aspartate pool induced by BCATI, the levels of carbamoyl-aspartate, dihydroorotate, and uridine monophosphate (UMP), all intermediates of de novo pyrimidine biosynthesis, together with inosine monophosphate (IMP) from purine biosynthesis pathways, were significantly decreased in 786-O cancer cells and their metastatic derivatives (Fig. 4f, g). Consistently with these observations, treatment with BCATI impairs the proliferation of all renal cells (Fig. 4h). Based on the observation that BCATI treatment does not fully deplete the aspartate and glutamate pool in renal cancer cells (Fig. 4e), we evaluated the contribution of other metabolic pathways to the synthesis of these key metabolites. We observed that several metabolites downstream of glutaminolysis (a-ketoglutarate, glutamate, 2-hydroxyglutarate) are accumulated in ccRCC cells (Fig. 2d, e) and it has been shown that glutaminolysis and reductive carboxylation of glutamine contribute to the synthesis of aspartate in ccRCC tumors[14]. Therefore, to measure the contribution of these two

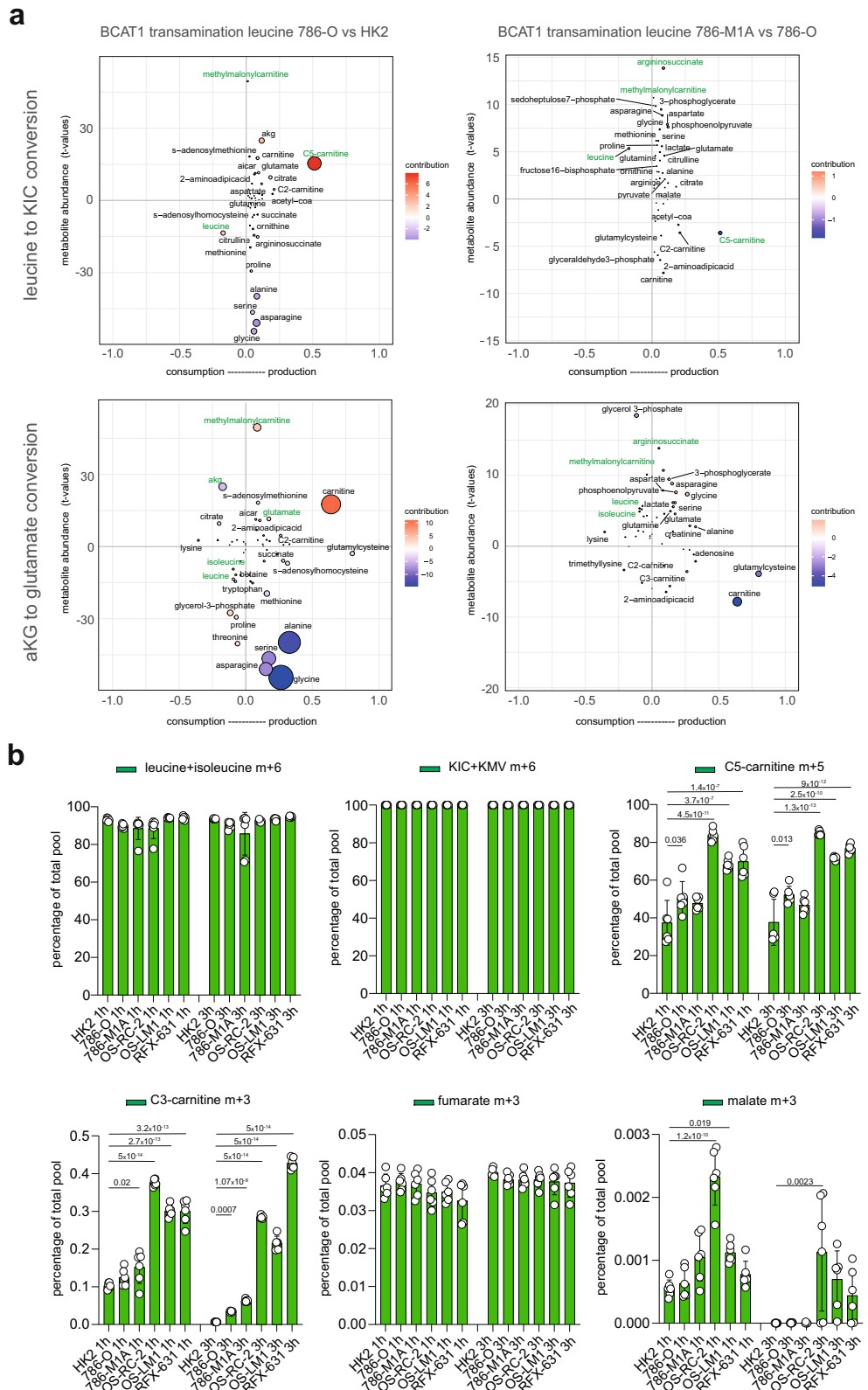

pathways to the generation of glutamate and aspartate in ccRCC cells in Plasmax, we performed a $^{13}C5$ glutamine tracing experiment in the presence of CB-839, a glutaminase inhibitor[39] and BMS-303141, a selective inhibitor of ATP Citrate Lyase[40] (ACLY, Supplementary Fig. 6a). As shown in Supplementary Fig. 6b, carbons from glutamine contribute to glutamate and aspartate carbons in all cell types, even though their contribution is reduced in cancer cells compared to HK2

(Supplementary Fig. 6b). Importantly, all cells displayed comparable percentages of citrate m+5, malate m+3, and aspartate m+3 (Supplementary Fig. 6b, c), suggesting a similar contribution of the reductive carboxylation of glutamine to the generation of these metabolites in all renal cells. As expected, the treatment of the cells with GLSI strongly suppressed the total pools of both aspartate and glutamate, comparably across cell lines (Supplementary Fig. 6d). Interestingly, treatment

**Fig. 3 | ocEAn, a tool to visualize metabolic changes in cancer cells.**
**a** Representative scatter plot generated using ocEAn for BCAT1 in the indicated comparisons. Metabolites upstream and downstream of BCAT1 directly or indirectly linked to reaction are indicated in two separate plots, one (on top) for conversion of leucine in ketoisocaproic acid (KIC), the other (on the bottom) for the transamination of α-ketoglutarate (aKG) to glutamate. The dot size represents the multiplication of the *t*-value with the weighted distance index (distance index being the number of the *x*-axis). *y*-axis reports the *t*-value of the abundances for the metabolites indicated in BCAT1 footprint including if they are accumulated or depleted upstream or downstream. The most relevant metabolites are highlighted in green. For the scatter plot generation, the methylmalonylcarnitine+succinylcarnitine metabolite was annotated as methylmalonylcarnitine only.
**b** Proportion of total pool of the indicated labeled metabolites originating from $^{13}$C leucine + isoleucine in all renal cells at the indicated time points. Data represent the mean of 5 independent cultures ± SD. *p*-values were calculated using one-way ANOVA with multiple comparisons and indicated in the graph for the comparisons HK2 vs. other biological groups at the given time point.

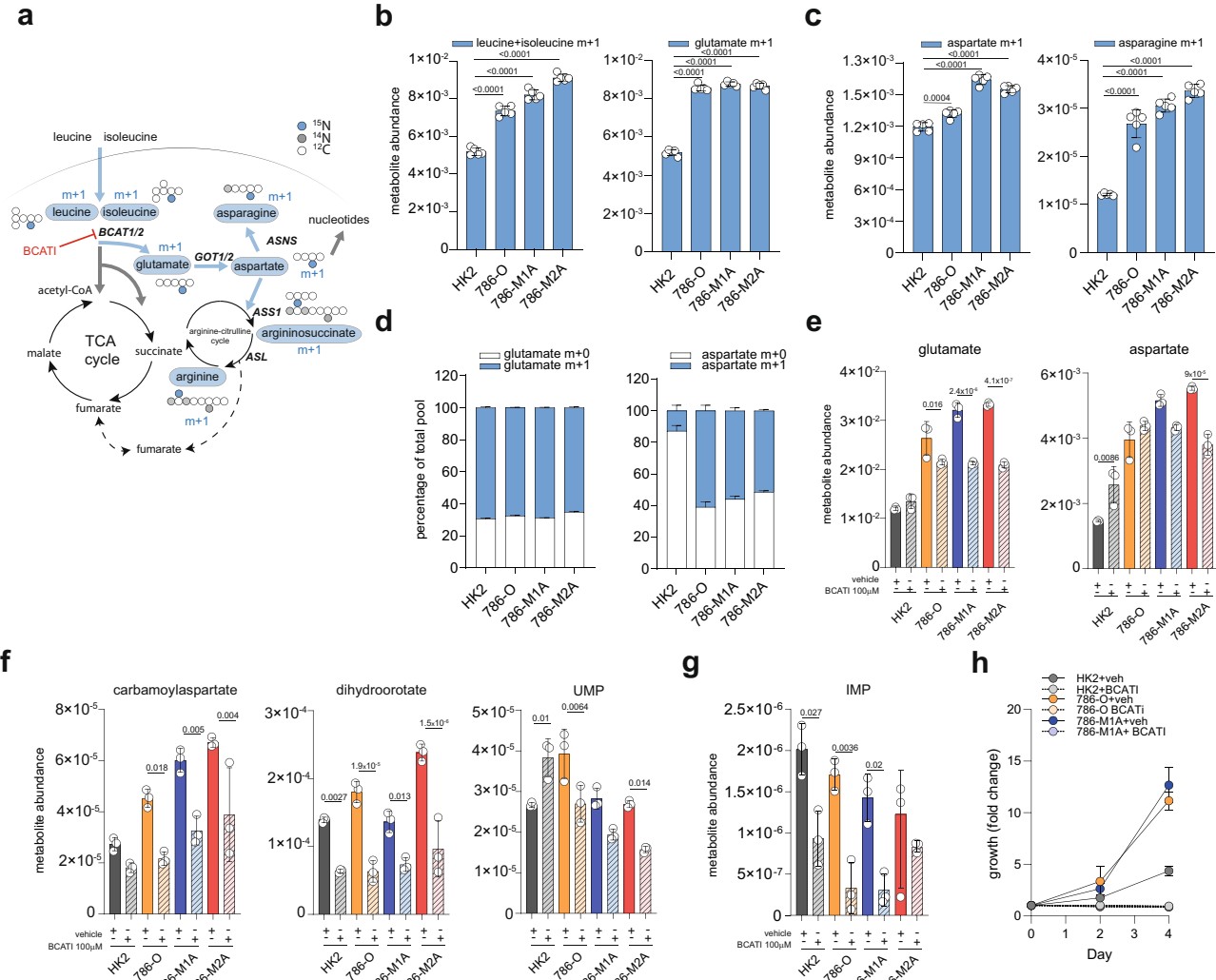

**Fig. 4 | BCAT transamination supplies nitrogen for aspartate and nucleotide biosynthesis in ccRCC. a** Diagram of the labeling pattern originating from $^{15}$N leucine catabolism. The gray circles indicate unlabeled N, blue circle $^{15}$N while white circles represent unlabeled carbons. Measured metabolites through LC–MS are indicated in blue circles. BCAT1/2 branched-chain amino acid transaminase 1/2, GOT1/2 glutamic-oxaloacetic transaminase 1/2, ASNS asparagine synthase, ASS1 argininosuccinate synthase, ASL argininosuccinate lyase. Abundance of labeled leucine m+1 and glutamate m+1 (**b**), aspartate m+1 and asparagine m+1 (**c**) originating from $^{15}$N leucine after 27 h normalized to total ion count. Data represent the mean of 6 independent cultures ± SD. *p*-values were calculated using one-way ANOVA where each group was compared with HK2. **d** Proportion of total pool of the indicated labeled metabolites originating from $^{15}$N leucine after 24 h in culture with EBSS + FBS 2.5% for 24 h. Data are normalized to total ion count and represent the mean of 6 independent cultures ± SD. **e–g** Intracellular abundance of the indicated metabolites after treatment with BCATI 100 μM in Plasmax for 22 h. Values are normalized to total ion count and expressed as the mean of 3 independent cultures ± SD. *p*-values were calculated using one-way ANOVA with multiple comparisons and indicated in the graph for the comparisons treated vs. vehicle for all biological groups. **h** Proliferation rate of the indicated cell lines in the presence of 100 μM of BCATI in Plasmax. Data represent the mean of 3 independent experiments ± S.E.M. Values represent fold-change increase of growth relative to day 0.

with ACLYI did not reduce asparate total levels across all renal cells even though the percentage of aspartate m+3 was halved in these conditions (Supplementary Fig. 6e, f), indicating that ACLY is not a major contributor of cytosolic aspartate generation in our conditions. Overall, these results indicate that while glutamine is an important source of the carbons of aspartate and glutamine, BCAA are essential nitrogen donours for these metabolites, especially in cancer cell lines.

We then investigated the molecular mechanisms underpinning the dysregulation of BCAA metabolism in ccRCC. *VHL* loss is a key driver in ccRCC formation, and through activation of HIFs, it is

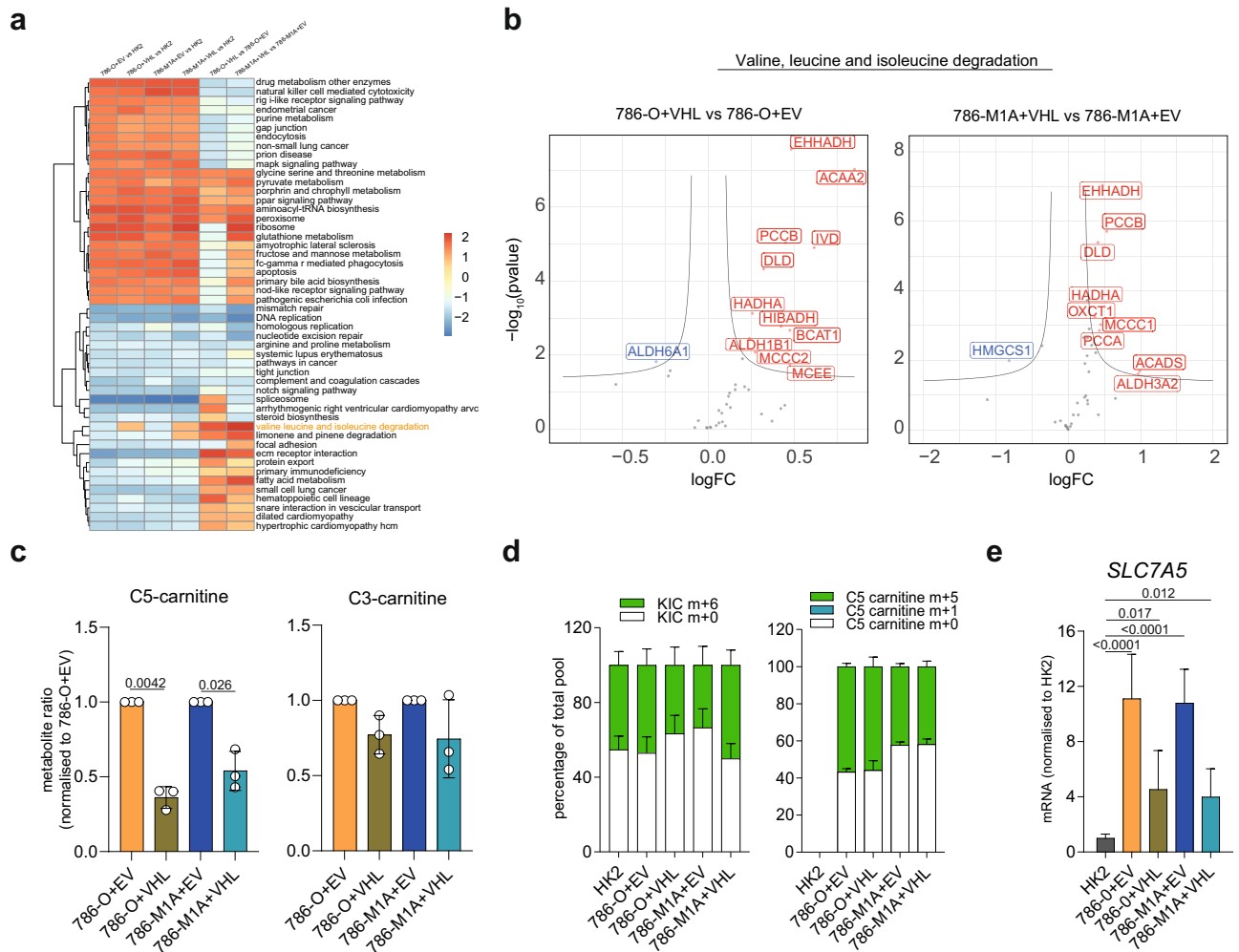

**Fig. 5 | VHL reconstitution restores BCAA functioning in ccRCC cells. a** Heatmap showing the enriched pathways in the indicated comparisons obtained through GSEA analysis of proteomics data generated from cells grown in RPMI. **b** Volcano plot showing the differential expression of proteins that belong to KEGG "Valine leucine and isoleucine degradation" signature in 786-O + VHL vs. 786-O + EV and 786-M1A + VHL vs. 786-M1A + EV from proteomics data obtained culturing cells in RPMI. FC fold-change, red = upregulated proteins, blue = downregulated proteins. **c** Ratio of the intracellular abundance of C3-carnitines and C5-carnitines in cells expressing VHL compared to EV cultured in RPMI. Data were normalized to total ion count and represent the mean of 3 independent experiments ($N = 3$) ± S.E.M. *p*-values were calculated using two-tailed one-sample *t*-test against the theoretical mean of 1 (786-O + EV = 1 vs. 786-O + VHL and 786-M1A + EV = 1 vs. 786-M1A + VHL). **d** Proportion of total pool of the intracellular ketoisocaproic acid (KIC) and C5-carnitine derived from $^{13}C$ leucine in renal cells cultured in RPMI. Data represents the mean of 5 independent cultures +SD. **e** mRNA levels of *SLC7A5* in the indicated cell lines grown in RPMI measured through qPCR. *TBP* was used as endogenous control. Values represent relative quantification (RQ) ± error calculated using Expression suite software (Applied Biosystem) calculated using SD algorithm. *p*-value was calculated through Expression suite software. $N = 3$ independent experiments.

responsible for the metabolic and bioenergetic reprogramming of renal cancer[14]. There is also evidence that under hypoxia, both HIF1A and HIF2A can transcriptionally regulate some of the genes of the BCAA catabolism such as *BCAT1* and *SLC7A5* in different tumor types[41–43]. Therefore, we investigated whether the rewiring of the BCAA catabolism in ccRCC depends on *VHL* loss. To address this question, we re-expressed wild-type *VHL* in 786-O and 786-M1A cells (Supplementary Fig. 7a). VHL expression restored mitochondrial respiration (Supplementary Fig. 7b) and increased aspartate level (Supplementary Fig. 7c). Next, we performed an enrichment analysis to identify which pathways are differentially regulated by VHL using additional proteomics data. Surprisingly, we found that BCAA catabolism is one of the most upregulated pathways in both 786-O and 786-M1A cells upon VHL restoration (Fig. 5a, b). As a consequence of the VHL-mediated transcriptional reprogramming, 786-O + VHL and 786-M1A + VHL cells showed significant suppression of C5 carnitines but not C3-carnitines accumulation (Fig. 5c), however no changes were observed in the C5-carnitines or KIC labelling patterns derived from $^{13}C_6$ leucine upon *VHL*

re-expression (Fig. 5d and Supplementary Fig. 7d). Moreover, we observed that VHL restoration induced almost 50% suppression of *SLC7A5*, the BCAA main transporter (Fig. 5e). Together with the re-expression of key proteins that belong to BCAA catabolism (Fig. 5b), the reduction of *SLC7A5* mRNA upon *VHL* re-expression confirmed that *VHL* loss is involved, at least in part, in the reprogramming of the BCAA degradation in renal cancer cells.

We then focused on the metabolic changes specific to the transition toward metastasis in the 786-O cellular model. As mentioned previously, ocEAn identified argininosuccinate as one of the key upregulated metabolites in metastatic cells compared to 786-O downstream of BCAT1 (Fig. 3a). Importantly, the nitrogen tracing experiments revealed the unexpected finding that the nitrogen from BCAAs was channeled into the biosynthesis of arginine through labeled aspartate, which is required to generate argininosuccinate by ASS1, in the metastatic 786-M1A and 786-M2A cells but not in normal HK2 or 786-O cells (Fig. 6a and Supplementary Fig. 5a). Of note, the abundance of labeled argininosuccinate is higher in the metastatic

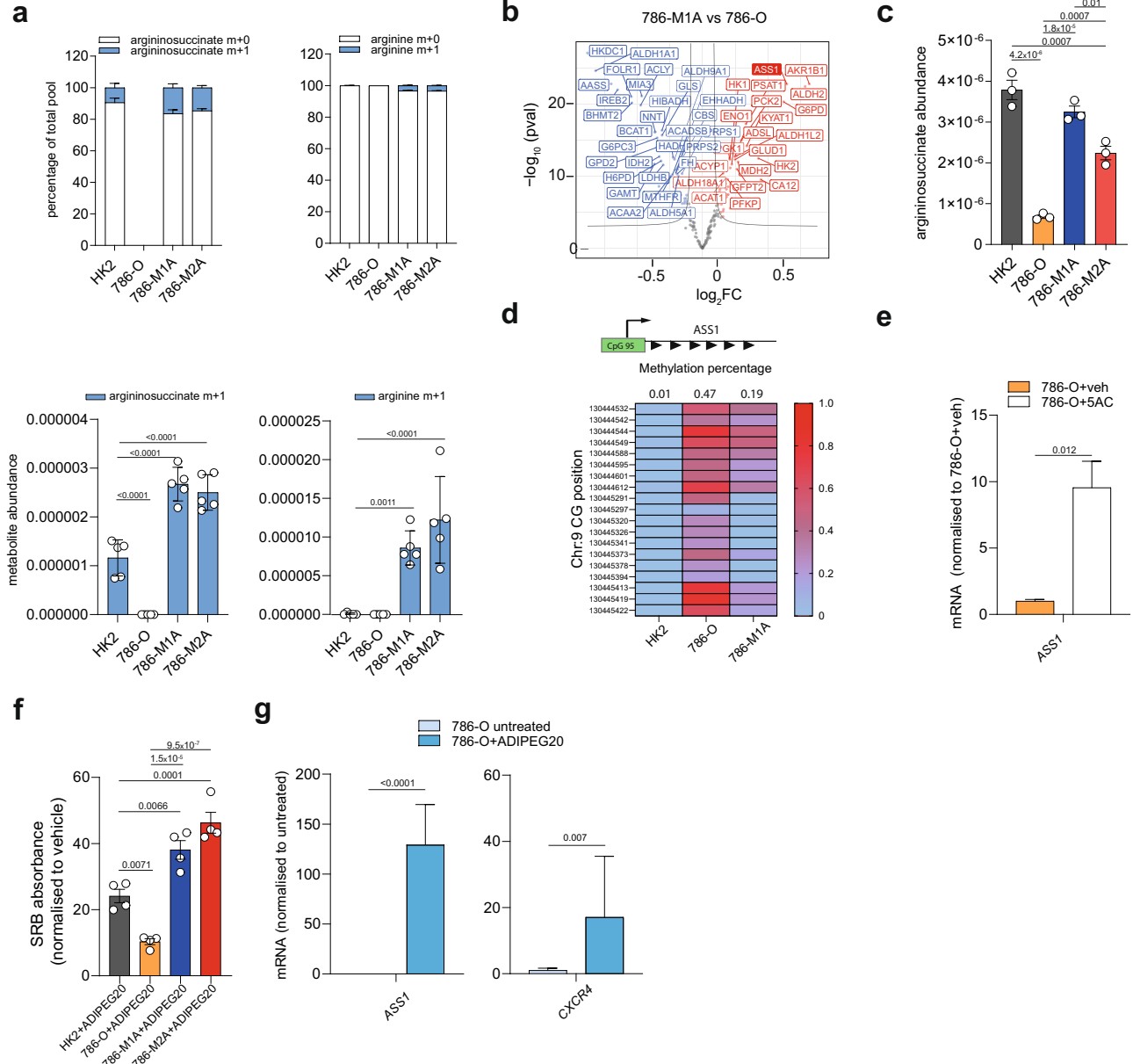

**Fig. 6 | ASS1 re-expression in metastatic ccRCC confers resistance to arginine depletion. a** Proportion of the total pool of the indicated labeled metabolites originating from $^{15}$N leucine (top) and abundance of labeled leucine m+1, glutamate m+1, aspartate m+1 (bottom) after 27 h normalized to total ion count. Data represent the mean of 6 independent cultures ± SD. *p*-values were calculated using one-way ANOVA where each group was compared with HK2. **b** Volcano of the differentially regulated metabolic genes comparing 786-M1A vs. 786-O using RNA-seq data generated from cells grown in Plasmax. red indicates upregulated genes, blue downregulated genes. FC fold-change. **c** Argininosuccinate abundance in the indicated cell lines cultured in Plasmax measured using LC–MS. Data were normalized to total ion count and represent the mean of 3 independent experiments ($N = 3$) ± S.E.M. *p*-values were calculated using one-way ANOVA with multiple comparisons. **d** Heatmap of the methylation level (*B*-value) of the indicated CGs within a CpG island overlapping with *ASS1* TSS. Values are presented as the mean of two independent experiments ($N = 2$). The average of all CGs methylation *b*-value is reported on top the heatmap for each cell type. **e** mRNA levels of *ASS1* in 786-O

treated for 72 h with either vehicle or 200 nM 5AC measured through qPCR. *TBP* was used as endogenous control. Values represent relative quantification (RQ) ± error calculated using Expression suite software (Applied biosystem) calculated using SD algorithm. *p*-value was calculated through Expression suite software. $N = 3$ independent experiments. **f** Measurement of cell proliferation through Sulforhodamine B (SRB) staining after treatment with pegylated arginine deiminase (ADI-PEG20, 57.5 ng/ml) at the indicated concentrations for 48 h. Values of SRB absorbance are shown as fold-change ± S.E.M. relative to vehicle-treated staining. *p*-values were calculated using one-way ANOVA with multiple comparisons. $N = 4$ independent experiments. **g** mRNA levels of *ASS1* and *CXCR4* in 786-O treated with ADIPEG20 57.5 ng/ml for 4 weeks, measured through qPCR. *TBP* was used as endogenous control. Values represent relative quantification (RQ) ± error calculated using Expression suite software (Applied biosystem) calculated using SD algorithm. *p*-value was calculated through Expression suite software from $N = 3$ independent experiments.

cells compared to HK2 (Fig. 6a). Since *ASS1* is known to be suppressed in 786-O and ccRCC[23], we hypothesized that *ASS1* might be reactivated in metastatic 786-M1A and 786-M2A cells. Accordingly, we detected higher ASS1 protein levels in metastatic cells compared to 786-O, with a mild increase in ASL levels even though not statistically significant,

while ARG2 was strongly suppressed, as shown before[23] (Supplementary Fig. 8a). To evaluate the specificity of ASS1 re-expression in the metastatic cells, we first focused on the metabolic genes differentially expressed between primary 786-O and metastatic 786-M1A cells cultured in Plasmax, using RNA-seq data (Fig. 6b). This analysis revealed

that *ASS1* was among the top upregulated metabolic genes in the metastatic cells together with Aldo-Keto Reductase Family 1 Member (*AKR1B1*), Aldehyde Dehydrogenase 2 (*ALDH2*) and Glucose-6-phosphate dehydrogenase (*G6PD*). We confirmed that ASS1 protein levels are restored in 786-derived metastatic cells using Western blot (Supplementary Fig. 8b) and that ASS1 expression was associated with increased intracellular levels of argininosuccinate in 786-M1A and 786-M2A cells (Fig. 6c) when compared to 786-O cells. Despite a differential regulation of *ASS1* among all the renal cell lines, we did not detect differences in the arginine uptake, except for 786-M2A cells, where it was considerably reduced compared to HK2. We also observed a higher release of ornithine in HK2 cells, while citrulline was selectively taken up only by the metastatic cells, which might be linked to ASS1 re-expression and its requirement for argininosuccinate biosynthesis (Supplementary Fig. 8c).

Next, we investigated how ASS1 expression was controlled in these cell lines. Based on previous reports showing hypermethylation of the *ASS1* promoter region and consequent gene suppression in different tumor types[44–49], we hypothesized that changes in methylation of *ASS1* promoter might control *ASS1* expression. Thus, we focused on a CpG island (hg38-chr9:130444478–130445423) that overlaps with the transcription starting site (TSS) of the gene (GRch38 chr9:130444200–130447801) and we measured its methylation using TruSeq Methyl Capture EPIC. We observed a gain of methylation at several CGs within this CpG island in 786-O, where *ASS1* is suppressed, while the same region is relatively hypomethylated in metastatic 786-M1A similarly to HK2 cells (Fig. 6d), where the gene is highly expressed. The treatment of 786-O with 5-azacitidine (5AC), a DNA demethylating agent, significantly increased *ASS1* expression, supporting the hypothesis that *ASS1* is epigenetically suppressed in primary renal cancer cells (Fig. 6e). Furthermore, we detected a strong peak of H3K27ac present at *ASS1* TSS in 786-M1A cells, which reflects the increased transcription of the gene (Supplementary Fig. 8d).

Next, we assessed if the reactivation of *ASS1* is a common phenomenon associated with the selection of metastatic cells by determining *ASS1* expression in the metastatic counterparts of OS-RC-2, the OS-LM1 cells, which were generated previously[27] (Fig. 2a). However, in this different metastatic model, *ASS1* mRNA is marginally upregulated in metastatic OS-LM1 compared to OS-RC-2 (Supplementary Fig. 8e) even though ASS1 was strongly suppressed in both ccRCC cells when compared to HK2 cells at the protein level (Supplementary Fig. 8f). Similarly to 786-O, the CpG island overlapping with the *ASS1* TSS is strongly hypermethylated in OS-RC-2, although we did not observe any change in its methylation levels in OS-LM1 (Supplementary Fig. 8g). We confirmed that ASS1 is epigenetically controlled by methylation in these cells since the treatment with 5AC leads to the re-expression of the gene in both cell lines (Supplementary Fig. 8h). Together, these data suggested that *ASS1* is epigenetically controlled in some but not all metastatic renal cancer cells. Therefore, *ASS1* upregulation might be present in a portion of advanced ccRCC tumors. To further corroborate this hypothesis, we analyzed changes in *ASS1* expression in human tumors from the TCGA RNA-seq dataset. Based on *ASS1* expression, we identified a cluster of advanced ccRCC (*ASS1*^high around 10% of the total cohort of cancers from stage III + IV) in which *ASS1* is significantly upregulated compared to stage I + II tumors, consistently with the phenotype observed in 786-M1A cells (Supplementary Fig. 9a). Intriguingly, this group of tumors is characterized by distinctive metabolic phenotype (Supplementary Fig. 9b), including upregulation of glycine, serine, and threonine metabolism, aspartate and glutamate metabolism, and OXPHOS, which strongly diverged from *ASS1*^low stage III + IV tumors (Supplementary Fig. 9b). Finally, we measured the accumulation of argininosuccinate in a small cohort (N = 18) of primary ccRCC from patients that were metastatic at the time of diagnosis. Some of the these tumors showed an increase in argininosuccinate levels compared to matched healthy tissue

(Supplementary Fig. 9c), suggesting that *ASS1* expression in advanced ccRCC might be heterogeneous. To corroborate the hypothesis that *ASS1* expression might be heterogenous in ccRCC we analyzed single-cell data available from a cohort of ccRCC tumors from patients[17]. Intriguingly, as shown in Supplementary Fig. 9d, even though the overall expression of *ASS1* is reduced in the tumor epithelial cells (red compartment) compared to renal proximal tubuli (green compartment), a few clones positive for *ASS1* expression are present even within the tumor. These data suggest that the expression of *ASS1* we observed in metastatic ccRCC might be the result of clonal selection, potentially driven by environmental cues through tumor progression.

It has been proposed that the suppression of *ASS1* induces arginine auxotrophy, sensitizing cancer cells to arginine depletion[50,51]. Our results suggest that primary and metastatic cells may exhibit different sensitivity to arginine depletion. Consistently, we found that the metastatic 786-M1A and 786-M2A cells were resistant to arginine depletion using the pegylated arginine deiminase (ADIPEG20), as a consequence of the restoration of *ASS1* in these cells (Fig. 6f). Given that *ASS1* expression confers the cells with the ability to survive in the absence of arginine, we hypothesized that the depletion of arginine might regulate *ASS1* expression in the renal cancer cells. Therefore, we chronically treated 786-O cells, where *ASS1* is suppressed, with ADIPEG20. Initially, the cells stopped proliferating until some sub-clones, resistant to the treatment, started to emerge. *ASS1* expression was upregulated in this population at mRNA level, corroborating the hypothesis of a pro-survival role of *ASS1* when arginine is rate-limiting, accompanied by a significant increase in the expression of the metastasis mediator *CXCR4* (Fig. 6g). Based on these results, we hypothesized that during tumor progression, renal cancer cells might be exposed to microenvironments that differ in arginine content. To corroborate this hypothesis, we measured the arginine levels in different mouse organs and their tissue interstitial fluids, focusing on comparing the kidneys and the lungs, the organ colonized by the metastatic population. Strikingly, arginine levels were significantly reduced in both the tissue and the interstitial fluid in the lungs compared to the kidneys (Supplementary Fig. 9e), suggesting that differences in arginine availability might directly contribute to the selection of metastatic subpopulations that re-express *ASS1* in the lungs.

Finally, we assessed whether ASS1 re-expression contributes to the metastatic features of this cell line. We observed that in 786-O cells treated chronically with ADIPEG20, *ASS1* increased expression was associated with *CXCR4* increase (Fig. 6g). On the other hand, silencing of *ASS1* (Fig. 7a) did not affect the proliferation of 786-M1A (Fig. 7b), but it strongly impaired the invasive growth of spheroids in collagen I matrixes, indicating that *ASS1* is required for the invasion and migration of metastatic cells in vitro (Fig. 7c). Based on the resistance to arginine depletion and the effects of *ASS1* silencing in vitro, we tested its effect on metastatic colonization and survival in vivo. Indeed, when injected into the tail vein of immunocompromised mice, 786-M1A + shASS1#1 and 786-M1A + shASS1#2 cells lost the ability to generate metastasis in the lungs (Fig. 7d–g). Thus, we confirmed that *ASS1* expression is necessary for ccRCC cells to maintain their invasiveness in vitro and colonize the lung in vivo.

Based on the high contribution of glutaminolysis to both glutamate and asparatate pools (Supplementary Fig. 6) and the key role of the aspartate levels for the growth of tumors, especially under hypoxia[37], we evaluated also if glutaminolysis contributes to invasion of ccRCC in vitro, measuring the invasive growth of spheroids derived from all cell types in the presence of GLSI and ACLYI. Even though we observed that both compounds significantly suppress the proliferation of all renal cells (Supplementary Fig. 10a, c), intriguingly, they did not impair the invasion of 786-M1A cells (Supplementary Fig. 10b, d). These results suggest that growth and invasion may be supported by distinct metabolic programs.

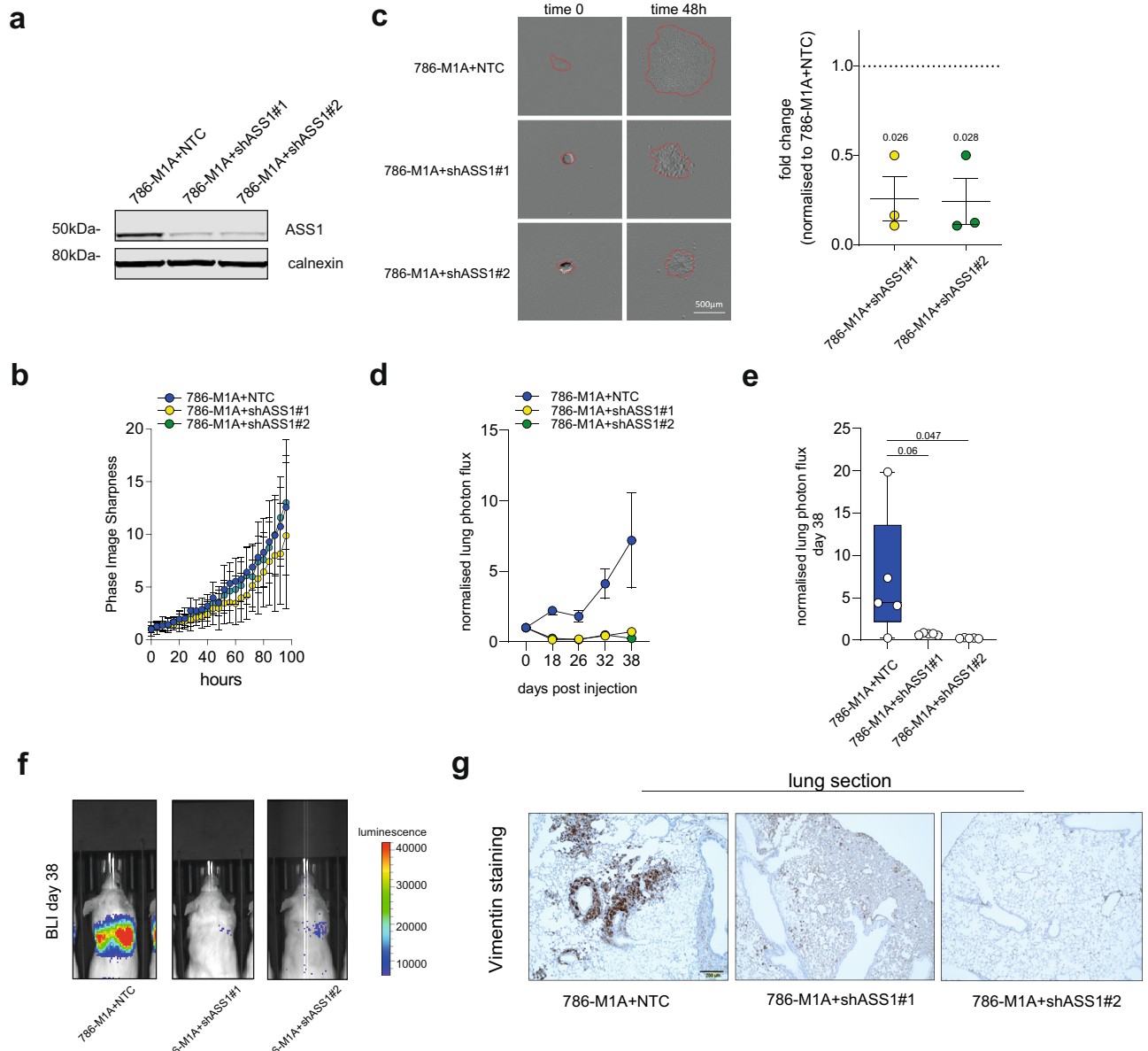

**Fig. 7 | ASS1 supports metastatic invasion in vitro and in vivo. a** Western blot of the ASS1 levels in cells stably cultured in Plasmax upon *ASS1* silencing using two different shRNA constructs. Calnexin was used as an endogenous control. **b** Measurement of 786-M1A cell proliferation after silencing of *ASS1* using Incucyte. Confluency values are shown as phase image sharpness calculated through Incucyte software ± S.E.M. *N* = 3 independent experiments. **c** Representative images of the indicated cell lines at time 0 and after 48 h (left) upon growth as spheroids in collagen (area marked in red). Pictures were obtained from Incucyte. Scale bar is 500 µm. Quantification of the cell spreading area in the collagen matrix at 48 h is represented as mean ± S.E.M from *N* = 3 independent experiments. Statistical significance was calculated using two-tailed one-sample *t*-test (null hypothesis ratio = 1). **d** Normalized lung photon flux from the lungs of 5 mice post tail-vein inoculation of 300,000 cells for the indicated cell types. Data are shown as mean of 5 mice ± S.E.M. **e** Box plot of the normalized lung photon flux of 5 mice/group at day 38 post-inoculation from the experiment shown in **d** and representative bioluminescence images of the mice at day 38 (**f**). 25% percentile/median/75% percentile are: 2.174;4.403;13.59 for 786-M1A + NTC, 0.5860;0.7700;0.8300 for 786-M1A + shASS1#1, 0.1940;0.2200;0.2895 for 786-M1A + shASS1#2. Statistical significance was calculated using one-way ANOVA with multiple comparisons. **g** Representative images of human vimentin/hematoxylin immuno-histochemistry of mouse lung sections after inoculation of cells in the tail vein for the indicated cell types. Scale bar is 200 µm.

## Discussion

Metabolic reprogramming is a hallmark of cancer[4,52] and in recent years, there have been significant efforts to map the metabolic landscape of different tumor types[4,7]. However, how cancer cells gain metabolic flexibility and its biological impact through tumor evolution is still largely unknown.

In this study, we exploited a panel of cell lines, including renal cells, tumoral and their metastatic derivatives, cultured under physiological nutrient conditions to model the metabolic phenotype of renal cancer through its progression. Of note, even though we used

only one normal epithelial renal cell line, HK2 cells are an an established cell line used in the majority of renal cancer studies and their metabolic phenotype closely resembles that normal renal tissue. Using this approach, we identified BCAA catabolism as one of the metabolic pathways strongly reprogrammed in renal cancer cells, whose transcriptional rewiring is sensitive to *VHL* restoration. Our findings are consistent with other works showing that hypoxia suppresses the BCAA catabolism in certain tissues[53] but upregulates the expression of *SLC7A5* and *BCAT1*[41–43] in different tumor types. By combining metabolomic labeling experiments and a novel computational tool (ocEAn),

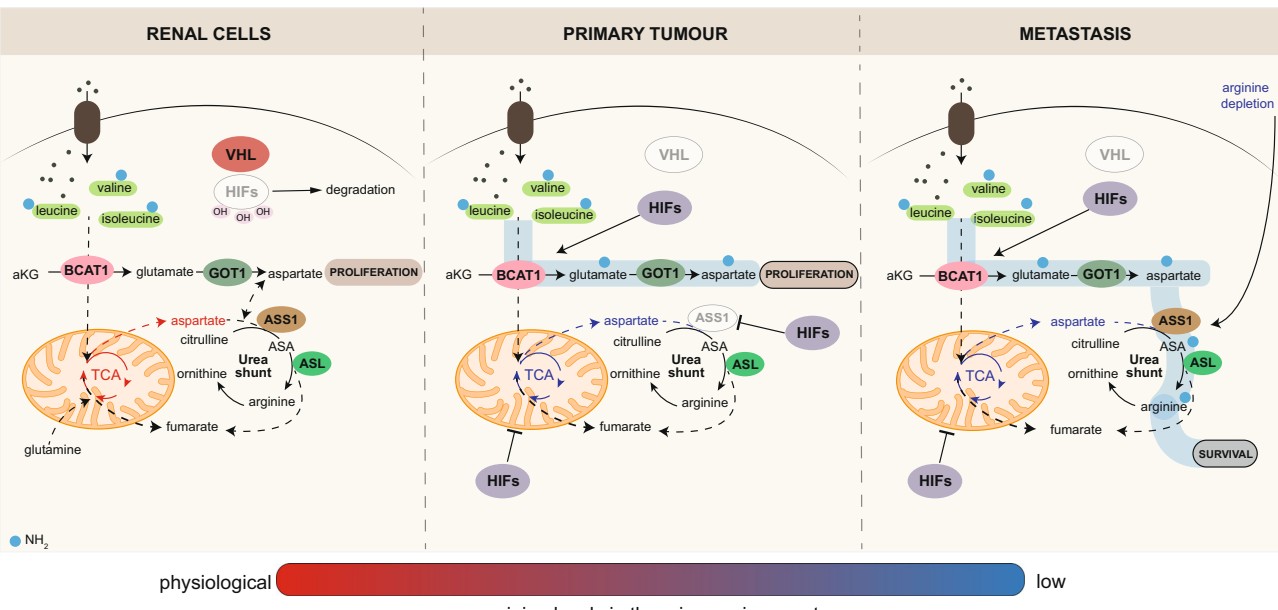

**Fig. 8 | Reprogramming of the BCAA amino acid catabolism is intertwined with the urea cycle enzymes during ccRCC progression.** Schematic showing a summary of the metabolic reprogramming in renal cancer cells during progression. Upon *VHL* loss, renal cancer cells activate a metabolic reprogramming to compensate for the aspartate defect that is a consequence of the HIF-dependent mitochondrial dysfunction present in these cells that involves combined activation of BCAT1 and GOT1. *ASS1* is suppressed, sparing aspartate from consumption through the urea cycle and favoring its re-direction towards nucleotide biosynthesis. In the metastatic population, *ASS1* is epigenetically reactivated, and its expression is triggered by low levels of arginine in the microenvironment. *ASS1* reactivation in the metastatic cells connects the BCAA catabolism reprogramming to the urea cycle, providing metastatic cells with the capability to derive arginine from BCAA and to survive in the presence of limiting levels of arginine.

we studied the regulation of the BCAA catabolism in renal cancer cells, demonstrating that it functions as a nitrogen reservoir for de novo biosynthesis of amino acids, especially aspartate and asparagine through BCAT transamination. Strikingly, under nutrient deprivation, renal cancer cells are capable to derive more than 60% of the aspartate nitrogen from BCAA transamination. Moreover, BCAT inhibition impairs the generation of intermediates of nucleotide biosynthesis, confirming other reports showing that BCAT is important for the proliferation[35] and the survival[38] of cancer cells. This metabolic rewiring is likely needed to compensate for the depletion of aspartate generated by the profound VHL-dependent mitochondrial defect observed in these cells. The epigenetic suppression of *ASS1* could be an additional metabolic strategy to spare aspartate and divert it to nucleotide biosynthesis in renal cancer cells as previously shown[54]. We also showed that other dysregulated metabolic pathways in ccRCC tumors such glutaminolysis supply carbons for the biosynthesis of aspartate further sustaining the proliferation of renal cancer cells under physiological conditions. However, inhibition of this pathway does not impair invasion of the cancer cells at least in vitro.

Our study showed that part of the nitrogen derived from BCAA is channeled into arginine biosynthesis only in the metastatic renal cells. The integration between the BCAA catabolism and urea cycle enzymes that emerged from our results bypassing the TCA cycle, was possible because of a selective epigenetic reactivation of the argininosuccinate synthase (*ASS1*) in the metastatic cells. This result was unexpected since it was recently shown that both *ARG2* and *ASS1* are frequently lost in ccRCC through copy number aberrations[23]. Our findings showed that at least in a fraction of ccRCC, *ASS1* is dynamically regulated and that its re-expression is necessary for ccRCC to retain full metastatic potential in vivo and in vitro. The evidence that *ASS1* is epigenetically silenced in other tumor types[55] and that arginine deprivation could trigger reactivation of *ASS1* in this condition[56] suggests that the re-expression of *ASS1* we observed in the metastatic renal cells might be driven by changes in arginine availability, an event that might have occurred at either the primary tumor level or the metastatic site. This hypothesis is corroborated by the evidence that the lung, one of the sites mostly colonized by ccRCC metastases, shows a lower level of arginine, both at tissue and interstitial fluid level, compared to the kidneys and that *ASS1* expression is heterogenous in primary tumors, with the detection of single-cell clones positive for *ASS1* even when *ASS1* is largely suppressed. Based on these results, a potential treatment of ccRCC patients with ADIPEG20, currently in clinical trials in a range of other cancer types (e.g., lung, liver and pancreatic cancers), should be carefully monitored since it might favor the selection of ASS1-proficient and more aggressive subpopulations from the primary tumor and could benefit from the combination of ASS1 inhibitors to target potentially metastatic clones.

In conclusion, we found that upon *VHL* loss, renal cancer cells activate a transcriptional rewiring that compensates for the suppression of the mitochondrial respiration and consequent depletion of aspartate through coordinated reprogramming of the BCAA catabolism and suppression of *ASS1* to sustain proliferation (Fig. 8). This mechanism is analogous to the activation of alternative metabolic routes to cope with a mitochondrial defect shown in different tumor types[57–61]. Finally, through tumor progression, the reactivation of *ASS1*, which is sensitive to the levels of arginine in the microenvironment and controlled by DNA methylation, provides the metastatic renal cancer cells with the selective advantage to channel nitrogen from BCAA to produce arginine when this aminoacid is scarce (Fig. 8). This metabolic flexibility is important for metastatic cells to survive in microenvironments with specific nutrient compositions and effectively colonize distant tissues.

## Methods
All experiments performed in the current study comply with all relevant ethical regulations.

## Cell culture

The human ccRCC 786-O and OS-RC-2 cells were obtained from J. Massaguè (MSKCC, NY-US) in 2014. Metastatic derivatives 786-M1A, 786-M2A and OS-LM1 have been previously described[27]. HK2 cells were a gift from the laboratory of Prof. Eamonn Maher (University of Cambridge, UK). All cells were authenticated by short tandem repeat (STR) and routinely checked for mycoplasma contamination and cultured in complete Plasmax medium prepared as described before[12] or RPMI (Sigma Aldrich) supplemented with 10% fetal bovine serum (FBS, Gibco-Thermo Scientific). All cells were cultured and passaged for at least 3 weeks in an incubator at 37 °C with 5% $CO_2$, to adapt to the Plasmax composition before starting to perform experiments. Counting for plating and volume measurement were obtained using CASY cell counter (Omni Life Sciences). Briefly, cells were washed in PBS once, then detached using trypsin-EDTA 0.05% (Gibco, Thermo Scientific).

## Cell proliferation measurement

Cell proliferation was analyzed using the Incucyte SX5 by means of phase contrast sharpness for 4 days, through sulphorhodamine B (SRB) staining or using crystal violet. For the SRB staining, $2 \times 10^4$ cells were plated onto 12-well plates (3 replicates/experimental condition for each cell line) and at each time point the cells were washed in PBS and incubated at 4 °C with 1% TCA solution. After two washes in water and once the plates were dry, the wells were treated with 0.057% SRB in acetic acid for 1 h at room temperature. After two additional washes in 1% acetic acid solution, plates were left to dry. To quantify the differences in the staining, the SRB was dissolved in 10 mM Tris solution and quantified using TECAN spectrophotometer reading the absorbance at 560 nm. Cell proliferation was quantified using crystal violet staining as previously described[62]. Briefly, $5 \times 10^4$ cells were plated onto 24-well plates (at least 3 replicates/experimental conditions for each cell line) and at each time point, cells were washed with PBS and fixed with 4% buffered formalin. Once all the time points were collected, the cells were washed with PBS and incubated with 0.1% crystal violet diluted in 20% methanol. After staining, the plates were finally washed with water and dried overnight. To quantify the staining, 0.5 ml 10% acetic acid was added to each well for 30 min at RT and the solution absorbance with the dye was quantified using TECAN spectrophotometer reading at 595 nm. For the treatment with inhibitors, cells were incubated with vehicle (DMSO) or BCATI 100 µM, GLSI 1 µM and ACLYI 10 µM, respectively.

## VHL re-expression in ccRCC cells

786-O and 786-M1A ± VHL cells were previously generated[29]. For comparison, cells transduced with empty vector (EV) were used. All cells were selected and then stably grown in RPMI with 2 µg/ml puromycin (Gibco, Thermo-Scientific).

## LC−MS metabolomics

**Steady-state metabolomics.** For steady-state metabolomics, $2 \times 10^5$ cells were plated the day before onto 6-well plates (5 or 6 replicates for each cell type) and extracted the day after. The experiment was repeated 3 times ($N = 3$). Before extraction, cells were counted using CASY cell counter (Omni Life Sciences) using a separate counting plate. After that, cells were washed at room temperature with PBS twice and then kept in a cold bath with dry-ice and methanol. Metabolite extraction buffer (MEB, 50% LC−MS grade Methanol (Fisher Scientific, 10284580), 30% LC−MS grade Acetonitrile (Fisher Scientific, 10001334) and 20% ultrapure water) was added to each well following the proportion $1 \times 10^6$ cells/1 ml of buffer. HEPES 100 ng/ml or Valine-d8 5 µM (CK isotopes, DLM-488) were used as internal standard for the MEB. After a couple of minutes, the plates were moved to the −80 °C freezer and kept overnight. The following day, the extracts were scraped and mixed at 4 °C for 10 min. After final centrifugation at max

speed for 10 min at 4 °C, the supernatants were transferred into LC−MS vials.

**Tracing experiments.** In all, $2 \times 10^5$ cells were plated onto 6-well plates (5 or 6 replicates for each cell type). The day after, the medium was replaced with fresh one containing the labeled isotopologue metabolite. For $^{13}C_6$ L-Leucine and $^{13}C_6$ L-Isoleucine (obtained from Cambridge Isotopes Laboratories) tracing experiment in Plasmax, cells were incubated for the indicated short time points or 43 h. For $^{15}N$ L-Leucine and Isoleucine (Sigma Aldrich) tracing for 27 h. The labeling experiment with $^{15}N$ L-Leucine in nutrient-deprived condition was conducted for 24 h in EBSS containing 2.5% FBS and 380 µM of $^{15}N$ L-Leucine (Sigma Aldrich). For the $^{13}C_5$ L-Glutamine (obtained from Cambridge Isotopes Laboratories) tracing experiments, the cells were cultured in Plasmax with 0.65 mM of labeled compound in the presence of vehicle, GLSI (CB-839, 100 nM) for 23 h or ACLYI (BMS-303141, 10 µM) for 8 h. For tracing experiments with RPMI, cells were incubated with $^{13}C_6$ L-Leucine, $^{13}C_5$ L-Valine or $^{15}N_2$ L-glutamine (obtained from Cambridge Isotopes Laboratories or Sigma Aldrich) for 24 h. In the manuscript, we presented the labeling patterns derived from $^{13}C_6$ Leucine for KIC and C5-carnitine in ccRCC cells + VHL, while only the total pools from labeling experiments have been used, together with other steady state experiments, to measure the intracellular levels of aspartate, C3 and C5 carnitines.

**Liquid chromatography coupled to mass spectrometry (LC−MS) analysis.** HILIC chromatographic separation of metabolites was achieved using a Millipore Sequant ZIC-pHILIC analytical column (5 µm, 2.1 × 150 mm) equipped with a 2.1 × 20 mm guard column (both 5 mm particle size) with a binary solvent system. Solvent A was 20 mM ammonium carbonate, 0.05% ammonium hydroxide; Solvent B was acetonitrile. The column oven and autosampler tray were held at 40 °C and 4 °C, respectively. The chromatographic gradient was run at a flow rate of 0.200 ml/min as follows: 0−2 min: 80% B; 2−17 min: linear gradient from 80% B to 20% B; 17−17.1 min: linear gradient from 20% B to 80% B; 17.1−22.5 min: hold at 80% B. Samples were randomized and analyzed with LC−MS in a blinded manner with an injection volume was 5 µl. Pooled samples were generated from an equal mixture of all individual samples and analyzed interspersed at regular intervals (every 6−8 samples) throughout the analysis of each experiment as a quality control (QC).

Metabolites were measured with a Thermo Scientific Q Exactive Hybrid Quadrupole-Orbitrap Mass spectrometer (HRMS) coupled to a Dionex Ultimate 3000 UHPLC. The mass spectrometer was operated in full-scan, polarity-switching mode, with the spray voltage set to +4.5 kV/−3.5 kV, the heated capillary held at 320 °C, and the auxiliary gas heater held at 280 °C. The sheath gas flow was set to 55 units, the auxiliary gas flow was set to 15 units, and the sweep gas flow was set to 0 unit. HRMS data acquisition was performed in a range of $m/z = 70−900$, with the resolution set at 70,000, the AGC target at $1 \times 10^6$, and the maximum injection time (Max IT) at 120 ms. Metabolite identities were confirmed using two parameters: (1) precursor ion m/z was matched within 5 ppm of theoretical mass predicted by the chemical formula; (2) the retention time of metabolites was within 5% of the retention time of a purified standard run with the same chromatographic method. Chromatogram review and peak area integration were performed using the Thermo Fisher software Tracefinder version 5.0. Only metabolites with less than 30% relative standard deviation in the QC samples are included. The peak area for each detected metabolite was then normalized against the total ion count (TIC) of that sample to correct any variations introduced from sample handling through instrument analysis. The normalized areas were used as variables for further statistical data analysis.

For $^{13}C$- and $^{15}N$- isotope tracing analysis, the theoretical masses of isotopes were calculated and added to a library of predicted isotopes.

These masses were then searched with a 5 ppm tolerance and integrated only if the peak apex showed less than 1% difference in retention time from the [U-$^{12}$C] monoisotopic mass in the same chromatogram. After analysis of the raw data, natural isotope abundances were corrected using the AccuCor algorithm (https://github.com/lparsons/accucor).

**Mouse tissue and interstitial fluid analysis.** Mice of a hybrid C57BL/6J;129/SvJ background were bred and maintained under pathogen free conditions at the MRC ARES Breeding Unit (Cambridge, UK). Animals of about 12 weeks of age were killed by neck dislocation and blood and tissues were speedily collected and processed for further analysis. Blood was recovered from the aorta, transferred to EDTA tubes (MiniCollect, Greiner Bio-One, 450531) and stored at −80 °C. The tissue samples were split into two: one snap-frozen in liquid nitrogen and stored at −80 °C until further processing, the second was used for interstitial fluid extraction using a protocol adapted from Sullivan et al. (2019)[63]. After the tissues were weighed, they were rinsed in room temperature saline (150 mM NaCl) and blotted on filter paper (VWR, Radnor, PA, 28298−020). We collected the hearts, livers, kidneys, and lungs from 8 wild-type mice and homogenized a piece of the tissue in metabolite extraction buffer using the proportion 25 μl/mg of buffer with Precellys Lysing tubes (Bertin Instruments). After that, extracts were kept in the freezer overnight and the following day centrifuged twice at max speed at 4 °C to remove the protein precipitates. Equal volumes of supernatants were spiked in with $^{13}$C arginine (Cambridge Isotopes) for quantification of arginine content. For extraction of the tissue interstitial fluid, we adapted the protocol from Sullivan et al. (2019)[63]. Briefly, the organ was washed in saline solution and then a portion was centrifuged at for 10 min at 4 °C at 106 x $g$ using 20 μm nylon filters (Spectrum Labs, Waltham, MA, 148134) affixed on top of 2 ml Eppendorf tubes. 1 μl of the eluate was extracted in 45 μl of extraction buffer and frozen overnight. The following day, all extracted were centrifuged twice at max speed at 4 °C to remove the protein precipitates. Supernatants were finally spiked in with $^{13}$C arginine (Cambridge Isotopes) for arginine quantification.

**Patient samples.** For the metabolic comparison in Supplementary Fig. 2a we used the data we generated and published in Dugourd et al. (2021)[9] from renal tumors and matched healthy tissue. The local ethics committee of the University Hospital RWTH Aachen approved all human tissue protocols for this study (EK-016/17). The study was performed according to the declaration of Helsinki. All patients gave informed consent. Kidney tissues were sampled by the surgeon from normal and tumor regions. The tissue was snap-frozen on dry-ice or placed in prechilled University of Wisconsin solution (#BTLBUW, Bridge to Life Ltd., Columbia, U.S.) and transported on ice.

The samples used for the metabolomics analysis in Supplementary Fig. 9c were generated using frozen tissue from surgically resected clear cell renal cell carcinoma samples that were sourced from an ongoing ethically approved study of biomarkers in urological disease (Ethics 03/018, CI V.J.G). From this study, we selected 18 primary tumors samples collected from patients treated from 2015–2017 and presenting with metastatic disease. Before processing the samples, whole frozen tissue areas of tumor were identified and marked by an uro-pathologist (A.Y.W). After that, samples were extracted for LC–MS analysis as described before[64].

**Consumption-release (CoRe) experiments.** $1.5 \times 10^5$ cells were seeded onto a 6-well plate and the experiment was carried as previously described[65]. Values represent the mean of five independent cultures ± S.D. and are relative to the metabolite abundance normalized to biomass dry weight generated (dW) in 24 h after medium background subtraction.

## RNA sequencing

In all, $4 \times 10^5$ cells were plated onto 5 replicate 6-cm dishes the day before the extraction. RNA isolation was carried using RNeasy kit (Qiagen) following the manufacturer's suggestions and the eluted RNA was purified using RNA Clean & Concentrator Kits (Zymo Research). RNA-seq sample libraries were prepared using TruSeq Stranded mRNA (Illumina) following the manufacturer's description. For the sequencing, the NextSeq 75 cycle high output kit (Illumina) was used and samples spiked in with 1% PhiX. The samples were run using NextSeq 500 sequencer (Illumina).

**Analysis.** Counts were generated from the read files using the Rsubread package with the hg38 genome build. Gene that had less than 50 counts per sample on average were filtered out. Then, 0 count values were scaled up to 0.5 (as done in the voom normalization procedure of the limma R package) and then log$_2$-transformed and normalized with the VSN R package. Differential analysis was then performed using the limma R package.

**Transcription factor activity from RNA-seq.** TF activities were estimated from the limma $t$-values as gene-level statistics, with the TF-target regulons from the dorothea v1.3.0 R package and the viper algorithm. Dorothea regulons were filtered to include TF-target interactions of confidence A, B, C and D. Viper was run with a minimum regulon size of 5 and eset filter set to FALSE. The resulting TF activity scores roughly represent how extreme is the average deregulation of a set of target genes of a given TF, compared to the rest of the genes.

## Proteomics analysis

**Sample preparation.** Cell pellets were lysed, reduced and alkylated in 100 μl of 6 M Gu-HCl, 200 mM Tris-HCl pH 8.5, 1 mM TCEP, 1.5 mM Chloractamide by probe sonication and heating to 95 °C for 5 min. Protein concentration was measured by a Bradford assay and initially digested with LysC (Wako) with an enzyme to substrate ratio of 1/200 for 4 h at 37 °C. Subsequently, the samples were diluted 10-fold with water and digested with porcine trypsin (Promega) at 37 °C overnight. Samples were acidified to 1% TFA, cleared by centrifugation (16,000 × $g$ at RT) and ~20 μg of the sample was desalted using a Stage-tip. Eluted peptides were lyophilized, resuspended in 0.1% TFA/water and the peptide concentration was measured by A280 on a nanodrop instrument (Thermo). The sample was diluted to 1 μg/5 μl for subsequent analysis.

**Mass spectrometry analysis.** The tryptic peptides were analyzed on a Fusion Lumos mass spectrometer connected to an Ultimate Ultra3000 chromatography system (both Thermo Scientific, Germany) incorporating an autosampler. 5 μL of the tryptic peptides, for each sample, was loaded on a homemade column (250 mm length, 75 μm inside diameter [i.d.]) packed with 1.8 μm uChrom (nanoLCMS Solutions) and separated by an increasing acetonitrile gradient, using a 150-min reverse-phase gradient (from 3% to 40% Acetonitrile) at a flow rate of 400 nL/min. The mass spectrometer was operated in positive ion mode with a capillary temperature of 220 °C, with a potential of 2000 V applied to the column. Data were acquired with the mass spectrometer operating in automatic data-dependent switching mode, with MS resolution of 240k, with a cycle time of 1 s and MS/MS HCD fragmentation/analysis performed in the ion trap. Mass spectra were analyzed using the MaxQuant Software package in biological triplicate. Label-free quantification was performed using MaxQuant. All the samples were analyzed as biological replicates.

**Data analysis.** Data were analyzed using the MaxQuant software package. Raw data files were searched against a human database (Uniprot Homo sapiens), using a mass accuracy of 4.5 ppm and 0.01 false discovery rate (FDR) at both peptide and protein levels. Every

single file was considered separately in the experimental design; the replicates of each condition were grouped for the subsequent statistical analysis. Carbamidomethylation was specified as fixed modification while methionine oxidation and acetylation of protein N-termini were specified as variable. Subsequently, missing values were replaced by a normal distribution (1.8 π shifted with a distribution of 0.3 π) in order to allow the following statistical analysis. Results were cleaned for reverse and contaminants and a list of significant changes was determined based on average ratio and t-test. Intensities were then normalized using the VSN package and differential analysis was performed with limma (same as for the RNA data). Gene set enrichment analysis was performed using the FGSEA package and the kegg pathway ontology (obtained from mSigDB).

### In vivo metastatic assay

All animal experiments were performed in accordance with protocols approved by the Home Office (UK) and the University of Cambridge ethics committee (PPL PFCB122AA). The housing for the animal experimental work carried out in this study was controlled by the animal facility at the University of Cambridge. Chow diet ad libitum was used during the experiments. For experimental lung metastasis assays, 300000 cells were resuspended in 100 μl PBS and inoculated in the lateral tail vein of 7-week-old female NOD/SCID mice (5 mice/condition) obtained from Charles River Laboratories. Metastatic colonization was monitored by IVIS bioluminescence imaging (PerkinElmer). At the experimental endpoint, lungs were harvested for immuno-histochemistry.

### Immuno-histochemistry staining (IHC)

Lungs were collected and fixed overnight with neutral formalin 4% and washed with PBS, 50% ethanol, and 70% ethanol for 15 min each. Lungs were embedded in paraffin, sectioned, and stained with H&E by the human research tissue bank and histopathology research support group from the Cambridge University Hospitals-NHS Foundation. Human Vimentin staining (Cell signaling #5741 1:100) was carried out using the Bond Max (Leica) using Bon polymer Refine Detection reagents (Leica) according to the manufacturer's protocol (IHC protocol F). Two different lung sections were vimentin-stained and imaged using Wide Field Zeiss Axio Observer 7 microscope (Zeiss).

### Oxygen consumption rate and extracellular acidification rate measurements

Cellular respiration (Oxygen consumption rate, OCR) was measured using the real-time flux analyzer XF-24e SeaHorse (Agilent) as described before[64]. Briefly, $6 \times 10^4$ cells were plated onto the instrument cell plate 24 h before the experiment in complete Plasmax medium or RPMI (at least four replicate wells for each cell line). The following day, the medium was replaced with fresh Plasmax supplemented with 25 mM HEPES (Sigma-Aldrich) to balance pH changes without any pre-incubation or with Agilent Seahorse XF RPMI with the addition of glucose, pyruvate and glutamine at the concentration present in normal RPMI and pre-incubated for 30 min at 37 °C. Cells were treated with 1 μM Oligomycin, 4 μM FCCP and 1 μM Antimycin A (all drugs were purchased from Sigma-Aldrich).

### TCGA KIRC transcriptomic analysis

KIRC RNA-seq counts were downloaded from the TCGA portal. Data were normalized in several steps. First, counts were $\log_2$ transformed. After visual inspection of the data distribution, any $\log_2$ count values lower than 7.5 were converted to missing values (NAs). Primary tumors and solid normal tissue samples containing more than 49000 NAs were removed. Then, genes with 350 or more missing values across samples were excluded. This yielded a clean data matrix of 593 samples and 13452 genes. The data was converted back to original count values so that VSN normalization procedure could be applied.

Groups were first defined as early-stage tumors (stage I and II) and late-stage tumors (stage III and IV). *ASS1* expression distribution was visually inspected in each group. Then the late-stage tumor group was split into two subgroups based on *ASS1* expression. We used Gaussian mixture modeling with the mclust package to model *ASS1* expression across late-stage tumor samples with two Gaussian distributions. This allowed us to define a group of low expression of *ASS1* (177 samples) and high expression of *ASS1* (18 samples, with a probability of sample belonging to a given distribution of 50%). Limma was used to perform differential analysis between late-stage tumors that express high/low *ASS1* and early-stage tumors. FGSEA (nperm = 1000) was used with the resulting limma t-values and KEGG pathway collection (obtained from msigdb) to perform a pathway enrichment analysis.

### Metabolomic enrichment analysis using ocEAn
#### Pre-processing of metabolomic data
**Steady-state metabolomics.** Three sets of metabolomic data relative to cells stably cultured in Plasmax were combined and the batch effect was removed with the removeBatchEffect function of limma (using a linear model to regress out the batch effect). We compared both 786-O and 786-M1A to HK2 and 786-M1A vs. 786-O using limma differential analysis and t-values relative to significant differences were calculated for each metabolite.

**Pre-processing of recon2 reaction network.** To run ocEAn, we first generated a list of metabolites associated with each enzyme. This information was extracted from the metabolic reaction network, indicating which metabolites are downstream or upstream of each reaction. The quality of the metabolic reaction network used to generate the set is of prime importance, as the choice of an adequate prior-knowledge source usually impacts the quality of footprint-based activity estimations the most. We used a reduced manually curated and thermodynamically proofed version of the Recon2 human metabolic reaction network to identify metabolites associated with each reaction[66]. The thermodynamic proofing was performed using the TFA algorithm to exclude reaction directions that were not thermodynamically feasible[67,68]. To compute the relative position of the metabolites relative to the enzymes, we first filtered out accessory elements of the reaction network such as cofactors and over-promiscuous metabolites (over-promiscuous metabolites are metabolites that are used as reactants by >100reactions). Metabolites classified as cofactors and nucleotides according to the KEGG BRITE classification were removed, as well as $CO_2$, ITP, IDP, NADH and all metabolites composed of less than four atoms. This procedure filtered out 100 metabolites, bringing the number of metabolites in the reaction network from 421 to 321.

**Convert redHuman network into an enzyme-metabolite distance map.** The gene-reaction rules "("AND" and "OR" which contains the information about which genes are required for a reaction to occur) of the metabolic reaction network were used to associate reactants and products with the corresponding enzymes of each reaction. When multiple enzymes were associated with a reaction with an "AND" rule, they were combined as a single entity representing an enzymatic complex. Then, reactants were connected to corresponding enzymatic complexes or enzymes by writing them as rows of a Simple Interaction Format (SIF) table in the following form: enzyme; 1; product. In this way, each row of the SIF table represents either activation of the enzyme by the reactant (i.e., the necessity of the presence of the reactant for the enzyme to catalyze its reaction) or activation of the product by an enzyme (i.e., the product presence is dependent on the activity of its corresponding enzyme). The resulting network allows to easily follow paths connecting metabolic enzymes with distant metabolites and can be converted to an enzyme-metabolite graph (using igraph package in R). The paths have to conserve the

compartment information of metabolites and reactions, thus enzymes and metabolites are duplicated and uniquely identified based on each reaction they are involved in. Finally, since the same enzyme catalyzes the transformation of different reactions (with variations of reactants and products), each reaction linked to a metabolic enzyme was uniquely identified (Supplementary Data 1). This level of resolution guarantees the correct tracking of a series of reactions from one metabolite to another without having incoherent jumps between metabolites catalyzed by the same enzyme.

The enzyme-metabolite graph was used to find the shortest path between each metabolic enzyme and all the other metabolites of the network. This is done first following the normal reaction fluxes (to connect enzymes with direct and indirect metabolic products) and then following the reversed fluxes (to connect enzymes with direct and indirect metabolic reactants). This yields a "reaction network forest", where each tree has a root corresponding to a specific metabolic enzyme, and branches represent the metabolites that can be reached from this enzyme, following normal or reverse reaction flux directions. Thus, each tree allows us to know if a given metabolite is upstream or downstream of a specific reaction and how many reaction steps separate them. The next step was associating each enzyme and all metabolites of the network with weights, representing the minimum distance of metabolites relative to enzymes, and a sign representing whether each metabolite is upstream ($-1$ to $0$) or downstream ($0$ to $1$) of a given enzyme. To compute a weight, we used a function that progressively decreases the weight value with the distance in the network. The weight value starts at 1 for direct reactants and products of a given enzyme and decreases in a stepwise manner ($x_{i+1} = x_i *$ penalty, with $x_0 = 1$ and dissipation parameter ranging between 0 and 1), for each reaction step separating the given metabolite from a given enzyme. In this study, we used a dissipation parameter of 0.8, which represents a drop in the weight of a given metabolite to a given enzyme by 20% per step in the reaction network. This value is arbitrary and was chosen because it allowed us to generate visually interpretable metabolic enzyme profiles. Since many cycles are present in the metabolic reaction network, metabolites are usually both upstream and downstream of different enzymes. To obtain a weight that represents the actual relative position of a metabolite relative to a given enzyme, the upstream and downstream weight of each metabolite-enzyme association were averaged.

### RNA extraction and real-time PCR
In all, $2.5 \times 10^5$ cells were plated onto a 6-well plate. The day after, cells were washed in PBS and then RNA was extracted using RNeasy kit (Qiagen) following the manufacturer's protocol. RNA was eluted in water and then quantified using Nanodrop (Thermo Fisher). 500 ng of RNA was reverse- transcribed using Quantitect Reverse Transcription kit. For real-time qPCR, cDNA was run using TaqMan™ Gene Expression Assay (FAM) (Thermo Scientific, catalog n. 4331182 250 µl, 20x: Hs00607978_s1 *CXCR4*; Hs03046964_s1 *VHL*; Hs01597989_g1 *ASS1*; Hs00427620_m1 *TBP*; Hs01001189_m1 *SLC7A5*) and Taqman Fast 2X master mix (Thermo Scientific). TATA-Box Binding Protein (*TBP*) was used as the endogenous control. Data and biological replicates were analyzed using Expression Suite (Thermo Scientific). Results were obtained from three independent experiments and presented as Relative quantification (RQ), with RQ max and RQ min calculated using SD1 algorithm. *p*-values were calculated by Expression Suite software.

### Treatment of cells with BCAT inhibitor
In all, $2 \times 10^5$ cells were plated onto 6-well plates (3 replicates/experimental condition for each cell line). The day after, media was replaced with fresh one with either the vehicle (DMSO) or 100 µM BCATI2, (ApexBio) for 22 h at 37 °C with 5% $CO_2$ before the metabolite extraction.

### Treatment of cells with DNA demethylation agent 5-azacitidine (5AC)
In all, $1 \times 10^5$ 786-O and OS-RC-2 cells were plated onto 6-well plates and incubated with either the appropriate vehicle or the inhibitor 5-Azacitidine, (Sigma-Aldrich) dissolved in DMSO at 200 nM concentration for 72 h 37 °C with 5% $CO_2$. The medium was replaced every day with fresh one containing either vehicle or the inhibitor. After a total of 96 h, cells were washed in PBS and RNA extracted as described above for real-time qPCR. The experiment was repeated three times ($N = 3$).

### Treatment of cells with ADIPEG20
In all, $3 \times 10^4$ cells were plated onto 24-well plates (4 replicates/conditions). The day after, pegylated arginine deiminase (ADIPEG20, Design Rx Pharmaceutical, US) was added at 115 ng/ml concentration for 72 h. Then, cells were fixed with 1%TCA solution at 4 °C for 10 min. After the plate was washed twice in water and dried, cells were colored using SRB staining solution (0.057% in acetic acid) for 1 h at room temperature. After two washes in 1% acetic acid solution and once dry, the SRB staining was dissolved in 10 mM Tris solution and absorbance quantified using TECAN spectrophotometer at 560 nm. The experiment was repeated four times ($N = 4$). For chronic treatment, 786-O cells were plated ($5 \times 10^5$) onto a T25 flask and treated with ADIPEG20 57.5 ng/ml for 4 weeks. Medium was replaced with fresh ADIPEG20 every 3 days.

### Short-hairpin RNA (shRNA) interference experiments
786-M1A were infected with lentiviral particles which were a gift from Ayelet Erez's laboratory. The virus was generated transfecting HEK293T cells with psPAX, pVSVG, which encode for the virus assembly, and pLKO shGFP, shASS1 vectors (Catalog #: RHS4533-EG445, GE Healthcare, Dharmacon). Cells were incubated with a medium containing the lentiviral particles for 24 h. After lentiviral transduction, cells were selected with puromycin 2 µg/ml for 48 h and then kept at 1 µg/ml for downstream experiments.

### Invasive growth assay
The invasive growth assay was performed as described previously[62,69]. Briefly, cells (1000 cells/drop) were maintained in drops (25 µl/drop) with Plasmax and 6% methylcellulose (Sigma M0387) on the cover of a 100-mm culture plate. Drops were incubated at 37 °C and 5% $CO_2$ for 72 h. Once formed, spheroids were collected, resuspended in collagen I solution (Advanced BioMatrix PureCol), and added to 24-well plates. After 4 h, Plasmax medium was then added on top of the well and day 0 pictures were taken. Any increase in spheroid area was monitored by taking pictures with Incucyte SX5 for 48 h or using the EVOS FL Auto Imaging system on days 0 and 2. For GLS and ACLY inhibitor treatments (CB-839, from Selleck Chemicals 1 µM and BMS-303141 from Merck 10 µM) cells were pre-treated for 48 h and 4 h, respectively, in the presence of the inhibitor or vehicle. For invasive growth quantification, an increase in the area occupied by the spheroids between day 0 and day 2 (48 h) was calculated using FiJi software.

### DNA methylation analysis
DNA samples (10 ng/µl, 500 ng total) were sheared using the S220 Focused-ultrasonicator (Covaris) to generate dsDNA fragments. The D1000 ScreenTape System (Agilent) was used to ensure >60% of DNA fragments were between 100 and 300 bp long, with a mean fragment size of 180–200 bp. The methylation analysis was performed using the TruSeq Methyl Capture EPIC Library Preparation Kit (Illumina) using the manufacturer's protocol. Twelve samples were pooled for sequencing on the HiSeq4000 Illumina Sequencing platform (single end 150 bp read) using two lanes per library pool. Technical replicates were performed for cell line data to assess assay reproducibility ($R^2 = 0.97$). The reads were trimmed (TrimGalore v0.4.4), aligned to the

bisulfite converted human reference genome (GRCh38/hg38) and methylation calling was performed using Bismark (v0.22.1) using standard parameters. Quality control (QC) reports were compiled using FastQC (0.11.4) and MultiQC (v1.7). The position of the CpG island (hg38-chr9:130444478–130445423) overlapping with the TSS of *ASS1* (GRch38 chr9:130444200–13044780) was obtained from *Ensembl*.

## Chromatin immunoprecipitation and sequencing (ChIP-seq)
ChIP experiments were generated and described previously[29].

## Protein lysates and western blot
In all, $6 \times 10^5$ cells were plated onto 6-cm dishes. The day after, cells were washed in PBS and then lysed on ice with RIPA buffer (150 mM NaCl, 1%NP-40, Sodium deoxycholate (DOC) 0.5%, sodium dodecyl phosphate (SDS) 0.1%, 25 mM Tris) supplemented with protease and phosphatase inhibitors (Protease inhibitor cocktail, Phosphatase inhibitor cocktail 2/3, Sigma-Aldrich) for 2 min. Cells extracts were scraped and then sonicated for 5 min (30 s on, 30 s off) using Bioruptor sonicator (Diagenode) and the protein content was measured using BCA kit (Pierce) following the manufacturer's instructions. Absorbance was read using TECAN spectrophotometer at 562 nm. 30–50 μg of proteins were then heated at 70 °C for 10 min in Bolt Loading buffer 1x (Thermo Scientific) containing 4% β-mercaptoethanol. Then, the samples were loaded into 4–12% Bis-Tris Bolt gel and run at 160 V constant for 1 h in Bolt MES 1X running buffer (Thermo Scientific). Dry transfer of the proteins to a nitrocellulose membrane was done using IBLOT2 (Thermo Scientific) for 12 min at 20 V. Membranes were incubated in blocking buffer for 1 h (either 5% BSA or 5% milk in TBS 1x + 0.01 % Tween-20, TBST 1X). Primary antibodies were incubated in blocking buffer ON at 4 °C. Calnexin antibody was purchased from Abcam (ab22595, used at 1:1000 dilution), ASS1 from Abcam (ab124465 used at 1:500 dilution). The day after, the membranes were washed three times in TBST 1X and then secondary antibodies (conjugated with 680 or 800 nm fluorophores, IRDye® 800CW Goat anti-Mouse IgG cat. 926–32210; IRDye® 800CW Goat anti-Rabbit IgG cat. 926–32211; IRDye® 680LT Goat anti-Mouse IgG cat. 926–68020; IRDye® 680LT Goat anti-Rabbit IgG cat. 926–68021 purchased fromLI-COR) incubated for 1 h at room temperature at 1:2000 dilution in blocking buffer. Images were acquired using Image Studio lite 5.2 (Li-Cor) on Odyssey CLx instrument (LI-COR).

## Analysis of ccRCC single-cell RNA-seq dataset
A single-cell ccRCC dataset was obtained from: https://doi.org/10.1126/science.aat1699.

The data of three ccRCC patients were analyzed with the scanpy python package. We filtered and normalized the data following the default parameters of the scanpy tutorial (filter out cells with less than 200 unique genes, filter out genes expressed in less than 3 cells, filter out cells with more than 5% mitochondrial gene counts). Data were summarized in lower dimensional space using UMAP dimensionality reduction. Finally, compartments (cell types, as defined in the original publication) and *ASS1* expression were plotted on the first two UMAP components. All the analysis codes are available in a markdown at: https://github.com/saezlab/marco_kidney_singlecell/blob/main/scripts/CCRCC_ASS1_final/CCRCC_ASS1_final.md

## Statistics and reproducibility
Graphs were generated using Graphpad Prism 8–9. The statistical analysis was performed using Prism software through either unpaired/paired *t*-test or one-way ANOVA using Dunnets's, Sidak's or Tukey's tests for multiple comparisons. For the one-way ANOVA analysis, gaussian distribution and equal SD were assumed. For real-time qPCR, the statistical analysis was performed using Expression Suite software

(Thermo Scientific) using SD algorithm on three independent experiments. The reproducibility of the experimental findings was supported by performing independent experiments (usually *n* = 3) or by having several independent culture replicates (replicate wells/dishes) as reported in the figure legends. Furthermore, additional experiments were conducted in other relevant cell lines to validate the main findings of the study. Western blot experiments were repeated more than once or validated with other techniques (qPCR or multi-omic analysis).

## Reporting summary
Further information on research design is available in the Nature Portfolio Reporting Summary linked to this article.

## Data availability
The RNA-seq raw files are available on Gene Expression Omnibus (GSE217211), while proteomic raw files are available on PRIDE (PXD037464). Metabolomics raw files are available on Metabolomics Workbench (project PR001418) and MetaboLights (study MTBLS5615).: All source data used to generate the figures presented in this study, including uncropped western blot images are provided in the Source Data file. Source data are provided with this paper.

## Code availability
All data and script for the analysis including ocEAn package are available at: https://zenodo.org/badge/latestdoi/395034170 and at https://github.com/saezlab/Sciacovelli_Dugourd_2021_paper. Whole network result visualization based on the ocEAn analysis of the metabolomics data is available at: https://sciacovelli2021.omnipathdb.org.

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

## Acknowledgements

We thank Dr. Alexandria Karcanias and Dr. Julien Bauer (Cambridge Genomic Services, Department of Pathology, University of Cambridge) for the RNA-seq library preparation and sequencing; the human research tissue bank and histopathology research support group from the Cambridge University Hospitals-NHS Foundation for the lung tissue processing. We thank Saverio Tardito for providing us with the Plasmax medium and for the guidance during the preparation of the medium formulation in house. We thank Denes Turei for helping set up the interactive metabolic networks online and Pau Badia for his help with exporting single-cell data from python to Excel. We thank all the members of Frezza's laboratory for critical reading and discussion of the manuscript. We thank Sivan Pinto for generating the shASS1 plasmids. A.E. is supported by research grants from the European Research Council (ERC 818943), and from the Israel Science Foundation (860/18). A.E. received additional support from the Moross Integrated Cancer Center, Koret foundation, Blumberg family, and from Manya and Adolph Zarovinsky. J.S.R. acknowledges funding by German Federal Ministry of Education and Research (Bundesministerium für Bildung und Forschung BMBF) MSCoreSys research initiative research core SMART-CARE (031L0212A) the European Union's Horizon 2020 research and innovation program to support A.D. (675585 Marie-Curie ITN "SymBioSys"). G.D.S. is supported The Mark Foundation for Cancer Research, the Cancer Research UK Cambridge Centre [C9685/A25177] and NIHR Cambridge Biomedical Research Centre (BRC-1215-20014). The views expressed are those of G.D.S. and not necessarily those of the NIHR or the Department of Health and Social Care. A.Y.W. is supported by the NIHR Cambridge Biomedical Research Centre and the Urological Malignancies Programme, funded by CRUK UK Major Centre Award C9685/A25117. V.G. acknowledges infrastructure support by the Cambridge Clinical Research Centre (NIHR) and Cambridge Biomedical Research Centre. L.V.J. is supported by a FEBS Long-term fellowship and by the CRUK Programme Foundation award to C.F. (C51061/A27453). C.S. work was funded by the European Union's Horizon 2020 research and innovation programme under the Marie Skłodowska-Curie grant agreement No 722605. M.S. and C.F. are supported by the MRC Core award grant MRC_MC_UU_12022/6.

## Author contributions

M.S. and C.F. conceptualized the study. M.S. designed and performed the majority of the experiments, interpreted the data and coordinated the research. M.S. and A.D. prepared the figures and M.S. wrote the manuscript with assistance of A.D. and all other authors. A.D. generated the computational tool ocEAn and performed all bioinformatic analyses supervised by J.S.R. and with the help of C.L., M.M., and V.H. L.V.J. generated the data relative to the silencing of ASS1, the in vitro invasion experiments with the metabolic inhibitors and performed the in vivo experiments with the help of V.C. and S.V. M.Y., E.N., A.S.H.C., and L.T. ran and analyzed the metabolomics samples. T.Y. and V.R.Z. collected the mouse tissue and extracted the samples for measurement of arginine in mouse tissue and interstitial fluids. P.R. generated ccRCC cells expressing wt-VHL and performed the ChIP-seq analysis relative to H3k27ac. D.G.R. provided advice and helped with the editing of the manuscript. C.S., S.H.R., and C.M. performed the EPIC methylation analysis and analyzed the data. A.E. provided reagents, advice and helped with editing of the manuscript. A.v.K. prepared the proteomic samples and analyzed the data. V.G., G.D.S., C.K., and R.K. collected the ccRCC patients used in the study. A.Y.W. performed the histopathological analysis of the patients' tumors. C.F. edited the manuscript and oversaw the research program.

## Funding

## Competing interests

G.D.S. has received educational grants from Pfizer, AstraZeneca, and Intuitive Surgical; consultancy fees from Pfizer, Merck, EUSA Pharma, and CMR Surgical; Travel expenses from Pfizer and Speaker fees from Pfizer. J.S.R. reports funding from GSK and Sanofi, and consultant fees from Travere Therapeutics and Astex Pharmaceutical. The remaining authors declare no competing interests.

## Additional information

[1]Medical Research Council Cancer Unit, University of Cambridge, Hutchison/MRC Research Centre, Box 197 Biomedical Campus, Cambridge CB2 0XZ, UK. [2]Department of Molecular and Clinical Cancer Medicine; Institute of Systems, Molecular and Integrative Biology, University of Liverpool, Liverpool L69 3GE, UK. [3]Faculty of Medicine and Heidelberg University Hospital, Institute for Computational Biomedicine, Heidelberg University, Heidelberg, Germany. [4]Institute of Experimental Medicine and Systems Biology, RWTH Aachen University, Aachen, Germany. [5]CECAD Research Center, Faculty of Medicine-University Hospital Cologne, 50931 Cologne, Germany. [6]Matterworks, Somerville, MA 02143, USA. [7]Early Detection Programme, CRUK Cambridge Centre, Department of Oncology, University of Cambridge, Hutchison/MRC Research Centre, Box 197 Biomedical Campus, Cambridge CB2 0XZ, UK. [8]Laboratory of Computational Systems Biotechnology, École Polytechnique Fédérale de Lausanne (EPFL), 1015 Lausanne, Switzerland. [9]Ludwig Institute for Cancer Research, Department of Oncology, Lausanne University Hospital (CHUV), University of Lausanne, CH-1011 Lausanne, Switzerland. [10]Division of Nephrology and Clinical Immunology, Faculty of Medicine, RWTH Aachen University, Aachen, Germany. [11]Edinburgh Cancer Research UK Centre, Institute of Genetics and Molecular Medicine, Crewe Road South, Edinburgh EH4 2XR, UK. [12]Department of Internal Medicine, Nephrology and Transplantation, Erasmus Medical Center, Rotterdam, The Netherlands. [13]Department of Surgery, University of Cambridge and Cambridge University Hospitals NHS Cambridge Biomedical Campus, Cambridge, UK. [14]Department of Histopathology-Cambridge University Hospitals NHS, Box 235 Cambridge Biomedical Campus, Cambridge CB2 0QQ, UK. [15]Department of Molecular Cell Biology, Weizmann Institute of Science, Rehovot, Israel. [16]Translational Cancer Medicine Program, Faculty of Medicine, University of Helsinki, Helsinki, Finland. [17]Department of Physiology, Faculty of Medicine, University of Helsinki, Helsinki, Finland. [18]These authors contributed equally: Marco Sciacovelli, Aurelien Dugourd. [19]These authors jointly supervised this work: Julio Saez-Rodriguez, Christian Frezza. ✉e-mail: pub.saez@uni-heidelberg.de; christian.frezza@uni-koeln.de

