## [Peer Review File · Nature Communications]

Reviewers' Comments:

Reviewer #1:

Remarks to the Author:

In the manuscript "Dynamic partitioning of branched-chain amino acids² derived nitrogen supports renal cancer progression" by Sciacovelli M. and colleagues the authors thoroughly describe how BCAA metabolism is regulated at different steps of ccRCC. Not only, they identify urea cycle as a crucial metabolic pathway in metastatic renal cancer cells which confers increased invasiveness in vitro and metastatization in vivo. The manuscript is well written, and the multipronged experiments are well conceived, making the evidence robust and consistent.

Nevertheless, I believe that there are some metabolic aspects that need to be analyzed more in depth:

1. The authors have well explained the role of BCAA catabolism in ccRCC. The suggestion to focus on this specific pathway is well described in Figure 1 and these results are validated in multiple cell lines in Figure 2-4, suggesting an important role of BCAAs as a nitrogen donor. On the other hand, in Figure 4e the authors clearly show that inhibition of transamination of BCAAs into glutamate is not sufficient to completely deplete glutamate and aspartate, as they decrease by 30% and 20%, respectively, in metastatic cell lines compared to HK2 cells.

Furthermore, in Figure 2d it was shown that α -ketoglutarate, 2-hydroxyglutarate, glutamate, and aspartate are all increased in 786-M1A/786-O compared with HK2, whereas glutamine, glutamate, and aspartate have higher levels in 786-M1A metastatic cells compared with 786-O. Together, these evidence may suggest an important contribution of the glutamine reductive pathway in ccRCC and ccRCC metastasis. In line with this, the role of ATP citrate lyase (ACLY) should be considered as it catalyzes the conversion of citrate to oxaloacetate and acetylCoA downstream of the glutamine reductive pathway in the cytoplasm. The authors should perform glutaminase I and ACLY inhibition experiments to quantify the contribution of BCAAs and glutamine to glutamate and aspartate biosynthesis, as well as L-glutamine-(amine-¹⁵N) and ¹³C⁵-glutamine tracing experiments.

2. The experiments performed in figure 6 do not causally link BCAA catabolism to cell invasion and metastasis, rather they clearly demonstrate that ASS1 is necessary for cell invasiveness. In line with this, the authors should functionally validate the invasiveness and the metastatization of shGLS1 (or shACLY) and shBCAT1 (or BCATI)-treated cells.

Minor points:

1. Fig. 3b: all panels are presented without statistical outcomes.
2. Fig. 4b and fig. 4c should be presented as fig. 4d.
3. Extended data fig.5d caption is missing.

Reviewer #2:

Remarks to the Author:

This is an exceptionally well conducted paper that demonstrates the nature of BCAA dysregulation in clear cell renal carcinoma. This reviewer tends to approach all of the old pathology names for cancers with caution, initially doubting that, for example, clear cell renal carcinoma is one thing. This body of work, however, showed not only that RCC is thing but that it its genetic particularities lead to particular and metabolic dependencies that enable and limit its proliferation in vitro and colonization of the lung in mice. My critiques are extremely minor. I prefer "dysregulation" to "deregulation" and would term azacytidine a demethylating agent as it doesn't exactly inhibit any DNMT as a small molecule but rather gets incorporated into DNA, trapping DNMT1 and resulting in DNMT1 proteolysis and net demethylation. The work ought to have a high impact and deserves publication in Nature Comms.

Reviewer #3:

Remarks to the Author:

In this report, Sciacovelli et al. identify suppression of BCAA as a feature of aggressive ccRCC based on analyses of publicly available transcriptomic data. They perform proteomic analyses that reveal BCAA enzyme suppression in a ccRCC cell line 786-O and a derivative from a metastasis compared to HK-2 proximal tubule cells. These data were extended to metabolomic analyses which showed lower levels of leucine/ isoleucine, and accumulation of C5 carnitines and methylmalonyl-carnitine, by-product metabolites of BCAA catabolism. They developed a computational method that generates a metabolic footprint for metabolic enzymes in a reaction network, ocEAn, showing that all ccRCC cells displayed lower BCAAs levels upstream of BCAT1 with a significant up-regulation of carnitines derived from intermediates. These experiments were complemented with tracing experiments using ¹³C6 leucine+isoleucine that showed no difference between HK2 cells and ccRCC cells and a limited contribution to the TCA cycle, as previously shown by others. They hypothesized that the reprogramming observed may provide a nitrogen source for glutamate. They performed tracing studies with ¹⁵N leucine+isoleucine and found that all ccRCC cells generated more glutamate, aspartate, and asparagine than HK2 cells. Then they cultured cells in medium devoid of aspartate/glutamate and in the presence of ¹⁵N leucine. They found that BCAT transamination to generate aspartate reached >60% in renal cancer cells, while in HK2 cells was <20% even though the intracellular percentage of labelled leucine was comparable. They did not observe, however, differences in the relative percentage of the labelled glutamate. Considering that aspartate is limiting for nucleotide biosynthesis, they assessed whether BCAT1 activity indirectly contributed to nucleotide pools. They suppressed BCAT1 with a pharmacological inhibitor, which reduced glutamate and aspartate levels. Using ocEAn, they identified argininosuccinate, a urea cycle intermediate produced by ASS1, as one of the key upregulated metabolites in metastatic cells compared to 786-O. Since ASS1 is known to be suppressed in 786-O, they hypothesized that ASS1 might be reactivated in metastatic 786-M1A and 786-M2A cells. They observed methylation at a CpG island in 786-O relatively to metastatic 786-M1A. They measured the accumulation of argininosuccinate in a small cohort of primary ccRCC from patients that were metastatic. Some of the metastatic tumors showed an increase in argininosuccinate levels compared to matched healthy tissue. They tested whether lower levels of ASS1 sensitized cancer cells to arginine depletion. Silencing ASS1 did not affect the proliferation of 786-M1A, but impaired growth of spheroids in collagen matrixes the ability to generate metastasis in the lungs

Main points

While the authors deserve to be commended for evaluating multiple ccRCC cell lines, they are all compared to HK-2, which is suboptimal as the results observed could simply be explained by particularities in HK-2 cells. Other 'normal' cell lines should be used to reproduce key results.

While ASS1 appears to be important for some metastatic ccRCC cell lines, this is not a consistent finding, which undermines the significance of the results.

Given the authors hypothesis, ASS1 should be examined in metastases, not primary tumors, even if giving raise to metastases.

They observed methylation at a CpG island in 786-O relatively to metastatic 786-M1A, but this was similar to HK2 cells, which expressed also high levels of ASS1 relative to 786-O.

Reviewer #5:

Remarks to the Author:

Authors present a detailed evaluation of the metabolomic reprogramming underpinning primary and metastatic clonally-related clear cell renal cancer cells (ccRCC) using a multi-omic (metabolome, proteome, transcriptome, epigenome) approach. Cell line models were cultured in PLASMAX, reflecting human serum nutrient composition, and branched chain amino acids (BCAA) catabolism was identified as a consistent metabolic pathway altered in primary and metastatic tumors compared to control HK2 cells (proximal tubule kidney epithelial cells). Authors further

found that BCAA provide the nitrogen required for de novo biosynthesis of aspartate and asparagine via BCAT transamination. This finding was uncovered through a novel metabolic-network based visualization approach developed as part of this work. Further evaluation of the transition from primary to metastatic cells identified that epigenetic activation of argininosuccinate synthase (ASS1), which is suppressed in primary ccRCC, modulates arginine availability, promotes invasion in vitro, and leads to metastasis in vivo. This study provides clinically relevant insight into ccRCC metabolic reprogramming that has strong translational research impact. Specifically, authors show that ADIPEG20, a pegylated arginine deiminase, restores ASS1 in metastatic cells, rendering them resistant to arginine depletion. Authors postulate that ADIPEG20 could thus favor selection of ASS1-proficient subclones and that treatment with ADIPEG20 and an ASS1 inhibitor combined could be more favorable.

Overall, this study sheds light on a BCAA-specific metabolomic reprogramming that provides a selective advantage to metastatic ccRCC cells. Results from ccRCC models are supported by in vivo work and evaluation of human tissues. The manuscript is very clearly written and organized, and the code and processed data (used for statistical analyses) are publicly available online.

Major Concerns:

- Lines 236-238, Extended Data Fig. 4c: Authors report an increase in glutamate, aspartate, and asparagine from BCAA compared to normal HK2 cells. Yet RFX-631 shows a decrease in labeled aspartate m+1 compared to control HK2 cells. This observation is contrary to the other ccRCC and metastatic cells tested and is not discussed.
- Lines 343-345, Extended Data Fig. 6k: Figure legend and methods indicate that argininosuccinate was measured in primary tissue, not metastatic tissue of metastatic ccRCC patients. Yet the text mentions "metastatic tumors" rather than primary tumors from metastatic patients. Also, were citrulline, arginine, and ornithine measured in these tumors? Did patterns mimic the cell line observations from Fig S6C?
- Some of the software versions and parameters are missing (e.g. Quan Browser Software, some packages, etc.).
- QC metrics on the reproducibility of the omic data is missing. For example, what was the agreement between biological replicates in the proteomic and metabolomic data? Unsupervised clustering analyses or other metrics could be shown as well.
- For the metabolomic data, details on study design and preprocessing are missing.
 - o There are no details on the study design for assessing robustness of measurements. Were pooled QCs run, and then used for batch effect correction? Does batch mean a different column was used for the LC? How reproducible are measurements across replicates?
 - o Parameters used for peak picking/validation are missing. Also, were any peaks filtered out based on low abundance/missing values?
 - o Details on identification of metabolites are missing. Was identification verified with standards (e.g. what level of identification is being reported?).
- The raw omic data (metabolomic, proteomic, methylation) newly measured for this study are not deposited in public repositories (e.g. Metabolomics Workbench, SRA, etc.).
- Extended Data Fig. 2a and lines 126-128: The heatmap does not show a clear difference in correlation between tumor and normal tissue metabolomic profiles for cells grown in Plasmax vs. RPMI. It is also unclear how the p-values were calculated and what exactly is being compared. The control HK2 Plasmax replicates seem more distinct compared to the ccRCC and metastatic derivative cells than RPMI. Also, the figure legend scale is a bit confusing since it is dichromatic. A monochromatic scale may be more easily interpretable here.

Minor Concerns:

- Figure 1A/C: the effect sizes are different or mislabeled (A is $-\log_{10}$ p-value while C is abs(NES) for the size of the black dots).
- Extended Data Fig. 2b and lines 128-130: it is unclear whether cells grown in RPMI express or not the expected transcription factors.
- It is unclear whether some experiments are performed on cells grown in Plasmax vs. RPMI (e.g. Line 138 for the proteomics data). If Plasmax was considered the most ideal growth media, then why were some experiments performed in RPMI (e.g. VHL-re-expression in ccRCC cells).
- Figure 2g: figure legend is missing.
- Line 162: authors write "BCAAs was substantially unaltered across all cell lines.". The phrase

"substantially unaltered" is not logical because one is not testing significance of "no change". Also, it seems that valine consumption is increased in 786-M1A relative to HK2.

- The network visualization available at <https://sciacovelli2021.omnipathdb.org> was not rendering well and due to lag time, I was not able to test the interactivity.

- Line 252: Extended Data Figure referenced should be 4e.

- Lines 272-273 and Fig. 5c: C3 carnitine in 786-M1A +VHL is not significantly suppressed compared to 786M1A as is stated in the text.

- Fig. 6a: Was argininosuccinate (native and labeled) measured in 786-O cells (top left plot in Fig. 6a)? It appears as if there was no data (rather than very low abundance).

- Lines 335-337: Are these observations from primary tissue? If so, please clarify.

- Line 804: how was correction for multiple comparisons performed?

- Line 810: the link to the public GitHub repo is wrong

Reviewer #1 (Remarks to the Author)

In the manuscript "Dynamic partitioning of branched-chain amino acids derived nitrogen supports renal cancer progression" by Sciacovelli M. and colleagues the authors thoroughly describe how BCAA metabolism is regulated at different steps of ccRCC. Not only, they identify urea cycle as a crucial metabolic pathway in metastatic renal cancer cells which confers increased invasiveness in vitro and metastatization in vivo. The manuscript is well written, and the multipronged experiments are well conceived, making the evidence robust and consistent.

We thank the referee for their positive comment and acknowledging the relevance and robustness of our manuscript.

Nevertheless, I believe that there are some metabolic aspects that need to be analyzed more in depth:

1. The authors have well explained the role of BCAA catabolism in ccRCC. The suggestion to focus on this specific pathway is well described in Figure 1 and these results are validated in multiple cell lines in Figure 2-4, suggesting an important role of BCAAs as a nitrogen donor. On the other hand, in Figure 4e the authors clearly show that inhibition of transamination of BCAAs into glutamate is not sufficient to completely deplete glutamate and aspartate, as they decrease by 30% and 20%, respectively, in metastatic cell lines compared to HK2 cells. Furthermore, in Figure 2d it was shown that α -ketoglutarate, 2-hydroxyglutarate, glutamate, and aspartate are all increased in 786-M1A/786-O compared with HK2, whereas glutamine, glutamate, and aspartate have higher levels in 786-M1A metastatic cells compared with 786-O. Together, these evidence may suggest an important contribution of the glutamine reductive pathway in ccRCC and ccRCC metastasis. In line with this, the role of ATP citrate lyase (ACLY) should be considered as it catalyzes the conversion of citrate to oxaloacetate and acetylCoA downstream of the glutamine reductive pathway in the cytoplasm. The authors should perform glutaminase I and ACLY inhibition experiments to quantify the contribution of BCAAs and glutamine to glutamate and aspartate biosynthesis, as well as L-glutamine-(amine-15N) and 13C5-glutamine tracing experiments.

We agree with the reviewer that understanding the net contribution of glutaminolysis and reductive carboxylation to the pool of glutamate and aspartate in renal cancer and metastatic cells is relevant for our study. We have already shown (Fig.4d) that when cultured in a medium without glutamine among other nutrients (EBSS+2.5% FBS with BCAA), the glutamate pool in all cells is highly dependent on the nitrogen derived from the BCAA. Intriguingly, for the aspartate pool, a high percentage of labelling from BCAA is observed only in the tumoral cells (higher than 60%, Fig.4d). These data suggest that the differential regulation of BCAT together with GOT transaminases contribute to the generation of aspartate in the cancer cells, at least in these experimental conditions, and further highlight the flexibility of renal cancer cells to differentially use substrates based on their availability for the synthesis of key amino acids.

To address the referee's request, we performed a $^{13}\text{C}_5$ glutamine tracing experiment in Plasmax in the presence of CB-839, a glutaminase inhibitor and BMS-303141, a selective

inhibitor of ACLY. Based on the results shown in Supplementary Fig. 6b, glutamine-derived carbons contribute substantially to the pool of aspartate and glutamate in all cell types, even though to a lower extent in the cancer cells compared to HK2 (higher percentage of m+0 is observed in all renal tumour cells+vehicle). This result is consistent with our findings that cancer cells can use different substrates to generate aspartate, such as BCAA. Of note, all cells show similar percentages of malate, aspartate (m+3), and citrate (m+5) derived from reductive carboxylation of glutamine (Supplementary Fig. 6a-b-c). As expected, GLS inhibition strongly reduces the total pools of both aspartate and glutamate in all renal cells (Supplementary Fig.6d). We did not observe major differences in the relative percentages of the isotopologues of aspartate and glutamate derived from $^{13}\text{C}_5$ glutamine after treatment with GLS inhibitor comparing the different cell types, with the percentage of m+0 being increased because of glutaminolysis inhibition (Supplementary Fig.6b). Intriguingly, ACLY inhibition did not reduce aspartate total levels, even though the percentage of the aspartate m+3 (as an indicator of reductive carboxylation) derived from $^{13}\text{C}_5$ glutamine is halved in all cell types (Supplementary Fig. 6a-e-f). These results show that glutaminolysis and reductive carboxylation significantly contribute to the carbon backbone of aspartate and glutamate pool. Importantly, however, this contribution does not vary across the cell lines tested. In addition, the observation that ccRCC cells display activation of the reductive carboxylation of glutamine and its biological impact have been extensively described in the literature both *in vitro* and in xenograft models (for instance see Gameiro et al.2013, Cell Metabolism). We added the new data and discussed the contribution of the glutaminolysis in the revised text (lines 265-288).

2. The experiments performed in figure 6 do not causally link BCAA catabolism to cell invasion and metastasis, rather they clearly demonstrate that ASS1 is necessary for cell invasiveness. In line with this, the authors should functionally validate the invasiveness and the metastatization of shGLS1 (or shACLY) and shBCAT1 (or BCAT1)-treated cells.

We thank the reviewer for their comments, which allow us to clarify an important point in our paper. First, in Fig. 6, we showed that the nitrogen derived from BCAA is used to generate both argininosuccinate and arginine in metastatic cells via ASS1, functionally connecting the two enzymes. At the same time, our data showed that one of the main biological outcomes of the BCAA-derived nitrogen reprogramming is to sustain survival through the generation of the aspartate, the substrate for both ASS1 and nucleotide synthesis, in both the metastatic and tumoral cells (Fig.4). Nevertheless, we agree that it would be important to directly measure the effect of BCAT inhibition on the invasive properties of renal cancer cells *in vitro*. Unfortunately, the pharmacological inhibition of BCAT enzymes with more prolonged incubation is detrimental to the proliferation of all cell types, including HK2 (see figure below and Fig.4h). Due to this marked effect on proliferation, we were not able to complete the spheroid formation assay. Finally, the pharmacological inhibition of both GLS and ACLY did not significantly affect the invasion of pre-formed spheroids from metastatic cells in collagen I matrixes, suggesting that even though glutaminolysis supports aspartate carbon backbone generation, its inhibition does not support the invasiveness of these cells *in vitro* (Supplementary Fig. 9b-d, lines 417-424).

Proliferation of the indicated cell lines in the presence of 100 μ M of BCATi (in Plasmax. Data represent the relative growth of cells at day 2-4 compared to day 0 from 3 independent experiments \pm S.E.M.

Minor points:

1. Fig. 3b: all panels are presented without statistical outcomes.
2. Fig. 4b and fig. 4c should be presented as fig. 4d.
3. Supplementary fig.5d caption is missing.

We thank the reviewer for all their comments, and we modified the text and figures accordingly. We showed now in the revised Fig.3b the percentages of the labelled metabolites derived from leucine+isoleucine carbons (m+3), including statistics significance (one-way ANOVA with multiple comparisons). We respectfully disagree on visualising the data of Fig.4b-c and Fig.4d in the same format because the point we are trying to make is a bit different. While in the case of Fig.4b-c we wanted to show the abundance of the labelled metabolites, in the experiment presented in Fig.4d, it was more relevant to show how the contribution of the nitrogen derived from leucine for the generation of aspartate is increased when other amino acids and nutrients are not available. Therefore, we feel that it is more informative the current representation for Fig.4d.

Reviewer #2 (Remarks to the Author)

This is an exceptionally well conducted paper that demonstrates the nature of BCAA dysregulation in clear cell renal carcinoma. This reviewer tends to approach all of the old pathology names for cancers with caution, initially doubting that, for example, clear cell renal carcinoma is one thing. This body of work, however, showed not only that RCC is thing but that it its genetic particularities lead to particular and metabolic dependencies

that enable and limit its proliferation in vitro and colonization of the lung in mice. My critiques are extremely minor. I prefer "dysregulation" to "deregulation" and would term azacytidine a demethylating agent as it doesn't exactly inhibit any DNMT as a small molecule but rather gets incorporated into DNA, trapping DNMT1 and resulting in DNMT1 proteolysis and net demethylation. The work ought to have a high impact and deserves publication in Nature Comms.

We would like to thank reviewer#2 for their very positive comments and for acknowledging the novelty and impact of our study. We replaced "deregulation" with "dysregulation" in the text and referred to azacytidine as a demethylating agent (lines 346; 793).

Reviewer #3 (Remarks to the Author)

In this report, Sciacovelli et al. Identify suppression of BCAA as a feature of aggressive ccRCC based on analyses of publicly available transcriptomic data. They perform proteomic analyses that reveal BCAA enzyme suppression in a ccRCC cell line 786-O and a derivative from a metastasis compared to HK-2 proximal tubule cells. These data were extended to metabolomic analyses which showed lower levels of leucine/ isoleucine, and accumulation of C5 carnitines and methylmalonyl-carnitine, by-product metabolites of BCAA catabolism. They developed a computational method that generates a metabolic footprint for metabolic enzymes in a reaction network, ocEAn, showing that all ccRCC cells displayed lower BCAAs levels upstream of BCAT1 with a significant up-regulation of carnitines derived from intermediates. These experiments were complemented with tracing experiments using ¹³C6 leucine+isoleucine that showed no difference between HK2 cells and ccRCC cells and a limited contribution to the TCA cycle, as previously shown by others. They hypothesized that the reprogramming observed may provide a nitrogen source for glutamate. They performed tracing studies with ¹⁵N leucine+isoleucine and found that all ccRCC cells generated more glutamate, aspartate, and asparagine than HK2 cells. Then they cultured cells in medium devoid of aspartate/glutamate and in the presence of ¹⁵N leucine. They found that BCAT transamination to generate aspartate reached >60% in renal cancer cells, while in HK2 cells was <20% even though the intracellular percentage of labelled leucine was comparable. They did not observe, however, differences in the relative percentage of the labelled glutamate. Considering that aspartate is limiting for nucleotide biosynthesis, they assessed whether BCAT1 activity indirectly contributed to nucleotide pools. They suppressed BCAT1 with a pharmacological inhibitor, which reduced glutamate and aspartate levels. Using ocEAn, they identified argininosuccinate, a urea cycle intermediate produced by ASS1, as one of the key upregulated metabolites in metastatic cells compared to 786-O. Since ASS1 is known to be suppressed in 786-O, they hypothesized that ASS1 might be reactivated in metastatic 786-M1A and 786-M2A cells. They observed methylation at a CpG island in 786-O relatively to metastatic 786-M1A. They measured the accumulation of argininosuccinate in a small cohort of primary ccRCC from patients that were metastatic. Some of the metastatic tumors showed an increase in argininosuccinate levels compared to matched healthy tissue. They tested whether lower levels of ASS1 sensitized cancer cells to arginine depletion. Silencing ASS1 did not affect the proliferation of 786-M1A, but impaired growth of spheroids in collagen matrixes the ability to generate metastasis in the lungs

Main points

While the authors deserve to be commended for evaluating multiple ccRCC cell lines, they are all compared to HK-2, which is suboptimal as the results observed could simply be explained by particularities in HK-2 cells. Other 'normal' cell lines should be used to reproduce key results.

We thank the reviewer for their comments. We agree that using only a normal cell line is a limitation of our study, which we acknowledged in the discussion of the revised manuscript (lines 430-435). We are aware of the availability of additional normal renal cell lines, but the HK2 remain the reference cell line for the majority of published studies because of its origin from the proximal renal tubuli, the site where ccRCC tumours are thought to develop. In addition, the other normal renal cell models are usually primary cells (while HK2 cells are immortalised) or if immortalised, they are grown in specific cell medium conditions, including specific supplements and growth factors that are often unknown to the user. We believe that all these limitations would negatively impact the comparison of additional normal renal cells and at the same time, be technically challenging. Finally, we believe the comparison between HK2 and the renal cancer cells used in the paper is relevant. Not only the metabolic phenotype of HK2 cells positively correlates with that of normal renal tissue (Supplementary Fig.2a) but also the GSEA of the pathways obtained from the comparison 786-O/786-M1A vs HK2 show similar results when compared with the analysis of the metabolic pathways dysregulated in the patients' samples (Fig.1A and Fig.2B). Finally, in the manuscript, we focused on the biological relevance of the re-expression of *ASS1* in metastatic 786-M1A cells compared to parental 786-O, which is independent of any control renal cell line such as the HK2 cells.

*While *ASS1* appears to be important for some metastatic ccRCC cell lines, this is not a consistent finding, which undermines the significance of the results. Given the authors hypothesis, *ASS1* should be examined in metastases, not primary tumors, even if giving raise to metastases.*

We thank the reviewer for their comments. We acknowledged that the selective re-expression of *ASS1* in metastatic renal cells that we uncovered is not a universal mechanism for all kidney metastases. That being said, we argue that we have found enough evidence to underline its prognostic and diagnostic value. As the reviewer pointed out, there are differences in the reactivation of *ASS1* between 786-O and OS-RC-2 cells, even though *ASS1* is silenced through epigenetic reprogramming in both cases (Fig.6d- and Supplementary Fig.8g). However, the two cell lines recapitulate what we observed in the transcriptional data from advanced ccRCC (Stage III/IV) present in TCGA, where we could identify two different clusters of tumours based on *ASS1* expression (Supplementary Fig.8i-j). We agree with the reviewer that directly analysing metastatic samples from patients or having access to databases from metastasis-only studies would improve our manuscript. Unfortunately, access to biopsies from patients' metastatic nodules or relevant datasets is limited and often does not provide enough resolution to clearly distinguish between cancer cells, fibrotic tissue derived from the target organs or immune infiltrates. For instance, this problem became apparent when we re-analysed three public datasets "GSE105288", "GSE85258", "GSE22541" (<https://peerj.com/articles/12493.pdf>). Indeed, when comparing the metastasis samples with primary kidney tumors in these studies, all

the most up-regulated genes were either related to lung tissue or fibrosis markers, indicating that a major part of the sample was contaminated by actually lung or fibrotic tissue. Considering these limitations and to further understand how *ASS1* expression is controlled in patients' renal tumours, we measured *ASS1* expression using a ccRCC tumours single-cell database previously published from <https://www.science.org/doi/10.1126/science.aat1699>. All analyses are available here: https://github.com/saezlab/marco_kidney_singlecell. The single-cell analysis showed that *ASS1* is overall silenced in most kidney tumour cells, but, importantly, we can consistently detect single clones with normal *ASS1* expression levels (Supplementary Fig.8I; lines 376-383). This new analysis suggests that the presence of *ASS1* in metastatic cells might depend on the different selection pressures tumour cells encounter through cancer evolution, including the availability of specific nutrients.

They observed methylation at a CpG island in 786-O relatively to metastatic 786-M1A, but this was similar to HK2 cells, which expressed also high levels of ASS1 relative to 786-O.

We apologise for the confusion generated by the colour-coded heatmap present in Fig.6d and Supplementary Fig. 8g. We now added an average methylation value for all the CGs detected in the CpG island overlapping with *ASS1* TSS on top of the graph. This value should better summarise the overall methylation changes of the region and be clearly consistent with the expression patterns of the gene in the cells.

Reviewer #5 (Remarks to the Author)

*Authors present a detailed evaluation of the metabolomic reprogramming underpinning primary and metastatic clonally-related clear cell renal cancer cells (ccRCC) using a multi-omic (metabolome, proteome, transcriptome, epigenome) approach. Cell line models were cultured in PLASMAX, reflecting human serum nutrient composition, and branched chain amino acids (BCAA) catabolism was identified as a consistent metabolic pathway altered in primary and metastatic tumors compared to control HK2 cells (proximal tubule kidney epithelial cells). Authors further found that BCAA provide the nitrogen required for de novo biosynthesis of aspartate and asparagine via BCAT transamination. This finding was uncovered through a novel metabolic-network based visualization approach developed as part of this work. Further evaluation of the transition from primary to metastatic cells identified that epigenetic activation of argininosuccinate synthase (*ASS1*), which is suppressed in primary ccRCC, modulates arginine availability, promotes invasion in vitro, and leads to metastasis in vivo. This study provides clinically relevant insight into ccRCC metabolic reprogramming that has strong translational research impact. Specifically, authors show that ADIPEG20, a pegylated arginine deiminase, restores *ASS1* in metastatic cells, rendering them resistant to arginine depletion. Authors postulate that ADIPEG20 could thus favor selection of *ASS1*-proficient subclones and that treatment with ADIPEG20 and an *ASS1* inhibitor combined could be more favorable.*

Overall, this study sheds light on a BCAA-specific metabolomic reprogramming that provides a selective advantage to metastatic ccRCC cells. Results from ccRCC models are supported by in vivo work and evaluation of human tissues. The manuscript is very clearly written and organized, and the code and processed data (used for statistical analyses) are publicly available online.

Major Concerns:

- Lines 236-238, Extended Data Fig. 4c: Authors report an increase in glutamate, aspartate, and asparagine from BCAA compared to normal HK2 cells. Yet RFX-631 shows a decrease in labeled aspartate m+1 compared to control HK2 cells. This observation is contrary to the other ccRCC and metastatic cells tested and is not discussed.

We thank the reviewer for this comment. We highlighted this discrepancy in the revised text (lines 241-244).

“Of note, a similar metabolic rewiring was observed in OS-RC-2, OS-LM1 and RFX-631, that derived a higher amount of glutamate, aspartate (with the exception of RFX-631 cells), and asparagine from BCAA when compared to normal HK2 cells (Supplementary Fig.5c)”

- Lines 343-345, Extended Data Fig. 6k: Figure legend and methods indicate that argininosuccinate was measured in primary tissue, not metastatic tissue of metastatic ccRCC patients. Yet the text mentions "metastatic tumors" rather than primary tumors from metastatic patients. Also, were citrulline, arginine, and ornithine measured in these tumors? Did patterns mimic the cell line observations from Fig S6C?

We thank the reviewer for this comment and apologise for the confusion. As the referee correctly pointed out, we measured the levels of argininosuccinate in primary tumours that were metastatic at the time of diagnosis. As discussed in the rebuttal point for reviewer#4, it is very difficult to access metastatic nodules from patients. Metastatic samples also present problems in interpreting the metabolomics results since they include lung tissue, immune infiltrate and fibrotic tissue. We indicated the origin of these samples in the figure panel (Supplementary Fig.8k) and changed lines 372-376 in the text. Finally, we also analysed the levels of citrulline, arginine and ornithine as suggested. As shown by the panel below, the profile of the other metabolites connected to the urea cycle seems to follow the same complex distribution of the ratios shown for argininosuccinate.

Ratio of the indicated metabolites of the urea cycle measured through LC-MS in a cohort of ccRCC patients' primary tumors that were metastatic at the time of diagnosis, normalized to healthy matched tissue. Red dots indicate ratio values >1, white <1.

- Some of the software versions and parameters are missing (e.g. Quan Browser Software, some packages, etc.).

Chromatogram review and peak area integration were performed using the Thermo Scientific™ Tracefinder™ software (Thermo Fisher Scientific, version 5.0) and the peak area for each detected metabolite was normalised against the total ion count of that sample to correct any variations introduced from sample handling through instrument analysis. The normalised areas were used as variables for further statistical data analysis.

- QC metrics on the reproducibility of the omic data is missing. For example, what was the agreement between biological replicates in the proteomic and metabolomic data? Unsupervised clustering analyses or other metrics could be shown as well.

We thank the reviewer for their suggestion. We included now additional figures for QC metrics for each omic dataset (Supplementary Fig.3, and below).

Supplementary Fig.3. Preliminary analyses (QC) of the multiomic datasets generated from renal cells cultured in Plasmax

- For the metabolomic data, details on study design and preprocessing are missing.

This information is included in the reporting summary. The metabolomic analysis was based on either performing independent experiments (usually n=3) or by having several independent culture replicates (replicate wells/dishes) as reported in the figure legends. Furthermore, additional experiments were conducted in other relevant cell lines to validate the main findings of the study. No sample size calculation was performed.

o-There are no details on the study design for assessing robustness of measurements. Were pooled QCs run, and then used for batch effect correction? Does batch mean a different column was used for the LC? How reproducible are measurements across replicates?

We use a pre-extraction internal standard valine-d8 to monitor sample preparation and stability of the LC-MS measurements. We prepare a pooled quality control sample by taking 10% from every sample in each experiment. The QC sample is injected inter-dispersed at regular intervals (every 6-8 samples) throughout the analysis of each experiment. Only experimental data with a less than 5% in relative standard deviation (RSD) of the internal standard valine-d8 are accepted. Additionally, all metabolites included in this study have an RSD of not more than 30% from the QC samples, while the metabolites with %RSD greater than 30% are excluded. The same column was used for all LC-MS experiments in this study. We use the raw data generated this way for downstream analysis. We do not perform batch correction to further manipulate the raw data. If the analysis from the pooled QC sample does not pass the criteria described above, we repeat the LC-MS run for the experiment. We hope this addresses the reviewer's question on the robustness and reproducibility of measurements.

- Parameters used for peak picking/validation are missing. Also, were any peaks filtered out based on low abundance/missing values? Details on identification of metabolites are missing. Was identification verified with standards (e.g. what level of identification is being reported?).

We have an in-house metabolite library of >1000 authentic compound standards. Metabolite identities were confirmed using two parameters: (1) precursor ion m/z was matched within 5 ppm of theoretical mass predicted by the chemical formula; (2) the retention time of metabolites was within 5% of the retention time of a purified standard run with the same chromatographic method. As described above, all metabolites included in the analysis must have RSDs of not more than 30%. Metabolites not meeting this standard were excluded.

- The raw omic data (metabolomic, proteomic, methylation) newly measured for this study are not deposited in public repositories (e.g. Metabolomics Workbench, SRA, etc.).

We will deposit all raw data to the appropriate public repositories after the acceptance of the manuscript.

- Extended Data Fig. 2a and lines 126-128: The heatmap does not show a clear difference in correlation between tumor and normal tissue metabolomic profiles for cells grown in Plasmix vs. RPMI. It is also unclear how the p-values were calculated and what exactly is

being compared. The control HK2 Plasmex replicates seem more distinct compared to the ccRCC and metastatic derivative cells than RPMI. Also, the figure legend scale is a bit confusing since it is dichromatic. A monochromatic scale may be more easily interpretable here.

We agree with the reviewer that the heatmap could have been better presented. We switched to a dichromatic scale and clearly highlighted the averaged correlation coefficient of the relevant comparisons that support the idea that plasmex medium allows for slightly better correlation with patient samples, while not changing dramatically from RPMI. Interestingly, this also highlighted that plasmex allows to more easily discriminate between primary tumor (786-0) and metastatic cell lines (786-M1A), since they cluster better than their RPMI counterpart. We updated the figure in Supplementary Fig.2a.

Minor Concerns:

- Figure 1A/C: the effect sizes are different or mislabeled (A is $-\log_{10}$ p-value while C is $abs(NES)$ for the size of the black dots).

Thank you for the comment, we updated the Fig. 1A/C and displayed the dot size as $-\log_{10}$ p-value together with the same colour code in both graphs.

- Extended Data Fig. 2b and lines 128-130: it is unclear whether cells grown in RPMI express or not the expected transcription factors.

We apologise for the confusion. Other studies previously published described the activation of relevant transcription factors in the same cell models grown in RPMI (for instance <https://doi.org/10.1038/nm.3029> and DOI: [10.1158/2159-8290.CD-17-1211](https://doi.org/10.1158/2159-8290.CD-17-1211)).

- It is unclear whether some experiments are performed on cells grown in Plasmax vs. RPMI (e.g. Line 138 for the proteomics data). If Plasmax was considered the most ideal growth media, then why were some experiments performed in RPMI (e.g. VHL-re-expression in ccRCC cells).

We thank the reviewer for their comment. All multiomic datasets (metabolomics, RNA-seq and proteomics) used in the manuscript were generated from cells cultured in Plasmax. Regarding the metabolomic profile of the cells re-expressing VHL, these are experiments we performed before starting to stably use the Plasmax formulation. RPMI is the most common medium used for ccRCC cells lines in the literature and the reprogramming of the BCAA (e.g accumulation of acyl-carnitines) is consistent between the two media (as can be seen also from the correlation heatmap comparing patient and cell line samples metabolomics in Supplementary Fig.2a). Therefore, we believe that using RPMI did not affect the results or their interpretation.

Figure 2g: figure legend is missing.

We apologise for this error. The figure legend is now updated.

- Line 162: authors write "BCAAs was substantially unaltered across all cell lines.". The phrase "substantially unaltered" is not logical because one is not testing significance of "no change". Also, it seems that valine consumption is increased in 786-M1A relative to HK2.

We apologise for the inaccuracy of this sentence, which was rephrased as:

"the uptake of BCAAs was not significantly different among the cells types, with the exception of valine in 786-O and 786-M1A cells" (lines 165-167)

- The network visualisation available at <https://sciacovelli2021.omnipathdb.org> was not rendering well and due to lag time, I was not able to test the interactivity.

We apologise for the inconvenience. The network webpage is quite large and can be difficult to display on some machines. While practical to generate interactive network webpages, the networkvis R package is not perfectly optimised.

- Line 252: Extended Data Figure referenced should be 4e.

We modified the text accordingly. Thank you for the suggestion.

- Lines 272-273 and Fig. 5c: C3 carnitine in 786-M1A +VHL is not significantly suppressed compared to 786M1A as is stated in the text.

We modify the text accordingly (lines 301-303), thank you for the comment.

“As a consequence of the VHL-mediated transcriptional reprogramming, 786-O+VHL and 786M1A +VHL cells showed significant suppression of C5 carnitines and C3 carnitines accumulation in 786-O+VHL only (Fig.5c)”

- *Fig. 6a: Was argininosuccinate (native and labeled) measured in 786-O cells (top left plot in Fig. 6a)? It appears as if there was no data (rather than very low abundance).*

It was below the limit of detection in these cells.

- *Lines 335-337: Are these observations from primary tissue? If so, please clarify.*

We apologise for the confusion. The cohort of samples analysed in Supplementary Fig. 8K are biopsies of primary tumours that are metastatic at the time of diagnosis. We edited the text (lines 372-376) and the figure panel accordingly.

- *Line 804: how was correction for multiple comparisons performed?*

Yes, FDR (Benjamini-Hochberg) was applied when appropriate for the transcriptomic, proteomic and metabolomic datasets.

- *Line 810: the link to the public GitHub repo is wrong*

Apologies for the confusion, the new corrected link is present below:
https://github.com/saezlab/Sciacovelli_Dugourd_2021_paper

Reviewers' Comments:

Reviewer #1:

Remarks to the Author:

The authors brilliantly answered all the concerns that this reviewer had raised. The work is therefore complete in the present form.

Reviewer #3:

Remarks to the Author:

The issues previously raised remain pertinent and are important, however, realizing the difficulties associated with their being addressed at this stage, and taking a broader perspective, I would be supportive of publication.

Reviewer #5:

Remarks to the Author:

Many thanks to the authors for carefully addressing all the comments.

Here are some remaining minor comments:

- The software/packages versions and references are still incomplete in the final version
- Authors appropriately answered the questions on the use of pooled QC for the metabolomic data and also the unsupervised analysis for QC of multi-omic data. These details should be incorporated in the manuscript as they are an important part of assessing data quality and providing clarity on best practices for the readership.
- Additional details on peak picking/validation provided in the response to reviewers should also be included in the main text. These details are essential for reproducibility/assessment.
- Thank you to the authors for planning on submitting the omic data to public repositories. Note however that public repositories typically allow submission of the data with an option to release the data publicly only when the publication is accepted. It is also possible to share these data with others (e.g. reviewers) during that time.

Reviewer #1

The authors brilliantly answered all the concerns that this reviewer had raised. The work is therefore complete in the present form.

We thank the reviewer for their positive comments and for helping us improving our manuscript.

Reviewer #3

The issues previously raised remain pertinent and are important, however, realizing the difficulties associated with their being addressed at this stage, and taking a broader perspective, I would be supportive of publication.

We thank the reviewer for their positive comments and for helping us improving our manuscript.

Reviewer #5

Many thanks to the authors for carefully addressing all the comments.

Here are some remaining minor comments:

- The software/packages versions and references are still incomplete in the final version

We thank the reviewer for their comments. We provided all details relative to the software and packages used at <https://zenodo.org/badge/latestdoi/395034170>.

Authors appropriately answered the questions on the use of pooled QC for the metabolomic data and also the unsupervised analysis for QC of multi-omic data. These details should be incorporated in the manuscript as they are an important part of assessing data quality and providing clarity on best practices for the readership. - Additional details on peak picking/validation provided in the response to reviewers should also be included in the main text. These details are essential for reproducibility/assessment.

We thank the reviewer for their comment. We added the additional details as requested in the methods session (Liquid chromatography coupled to Mass Spectrometry (LC-MS) analysis paragraph, lines 569-596).

Thank you to the authors for planning on submitting the omic data to public repositories. Note however that public repositories typically allow submission of the data with an option to release the data publicly only when the publication is accepted. It is also possible to share these data with others (e.g. reviewers) during that time.

We thank the reviewer for their comment. Proteomics and RNA-seq raw files are available at:

<https://zenodo.org/badge/latestdoi/395034170>. We submitted the proteomic original files to ProteomeXchange as well, where they are currently under reviewing. The files can be accessed using the following temporary parameters:

Project accession: PXD037464

Reviewer account details:

Username: reviewer_pxd037464@ebi.ac.uk

Password: ALHIJdoD

We apologize for the delay to make all raw metabolomic files publicly accessible. A part of the metabolomic datasets is already available at:

<https://www.metabolomicsworkbench.org/data/DRCCMetadata.php?Mode=Project&ProjectID=PR001418>. We provided access to the additional metabolomic raw files currently under reviewing on MetaboLights to the editor and the referees. Given the amount of the data to deposit, this process is also taking a longer amount of time compared to what we originally predicted, also because some of the experiments were run by different laboratory members over the time. We will provide the access code and reference as soon as the reviewing process is completed. In the meantime, all source data used to generate the graphs relative to the metabolomic analyses are available in the Source Data file. DNA Methylation percentages of CG shown in the graph at ASS1 TSS for all replicates are included in the Source Data File as well.